# Breaking photoswitch activation depth limit using ionising radiation stimuli adapted to clinical application

Alban Guesdon-Vennerie[1], Patrick Couvreur[1], Fatoumia Ali[1], Frédéric Pouzoulet[2,3], Christophe Roulin[2,3], Immaculada Martínez-Rovira [4], Guillaume Bernadat[5], François-Xavier Legrand [1], Claudie Bourgaux[1], Cyril Lucien Mazars[1], Sergio Marco [6], Sylvain Trépout [6], Simona Mura[1], Sébastien Mériaux [7] & Guillaume Bort [1,8✉]

Electromagnetic radiation-triggered therapeutic effect has attracted a great interest over the last 50 years. However, translation to clinical applications of photoactive molecular systems developed to date is dramatically limited, mainly because their activation requires excitation by low-energy photons from the ultraviolet to near infra-red range, preventing any activation deeper than few millimetres under the skin. Herein we conceive a strategy for photosensitive-system activation potentially adapted to biological tissues without any restriction in depth. High-energy stimuli, such as those employed for radiotherapy, are used to carry energy while molecular activation is provided by local energy conversion. This concept is applied to azobenzene, one of the most established photoswitches, to build a radioswitch. The radiation-responsive molecular system developed is used to trigger cytotoxic effect on cancer cells upon gamma-ray irradiation. This breakthrough activation concept is expected to expand the scope of applications of photosensitive systems and paves the way towards the development of original therapeutic approaches.

[1] Université Paris-Saclay, CNRS, Institut Galien Paris-Saclay, 92296 Châtenay-Malabry, France. [2] Institut Curie, PSL Research University, Translational Research Department, Experimental Radiotherapy Platform, UMR 1288, F-91405 Orsay, France. [3] Université Paris Sud, Université Paris-Saclay, Translational Research Department, Experimental Radiotherapy Platform, UMR 1288, F-91405 Orsay, France. [4] Ionizing Radiation Research Group, Physics Department, Universitat Autònoma de Barcelona, 08193 Bellaterra, Cerdanyola del Vallès, Barcelona, Spain. [5] Université Paris-Saclay, CNRS, BioCIS, 92290 Châtenay-Malabry, France. [6] Institut Curie, Université Paris-Saclay, CNRS UMR9187, INSERM U1196, 91405 Orsay, France. [7] Université Paris-Saclay, CEA, CNRS, BAOBAB, NeuroSpin, 91191 Gif-sur-Yvette, France. [8] University of Lyon, Université Claude Bernard Lyon 1, CNRS, Institut Lumière Matière, F-69622 Villeurbanne, France. ✉email: guillaume.bort@cnrs.fr

Triggering process activation using stimuli is a promising approach to reach control and modulation of on-demand therapeutic actions in real time[1]. Stimulus-triggered actions are mediated by endogenous or external stimuli. While the former restrict to specific local environments of the targeted zone, such as distinctive pH, redox potential, oxygen content or enzyme activity, the latter enable to escape from these local limitations thanks to an orthogonal actuation leading to high spatiotemporal control[2,3]. Due to the promising benefits of this approach, light responsive systems have been developed for many decades and are now able to initiate specific and complex actions such as bond cleavage, switch, slide or rotation, which led to the research field of photoactivation[4].

The first reported photosensitive systems were activated by ultraviolet (UV) photons carrying enough energy to induce bond cleavage or molecular motion. Because of the limited tissue penetration of such photons (<100 μm), many investigations were undertaken to reach deeper areas in the body by lowering the required energy of the activating photon[5]. New chemical entities, such as visible or near-infrared absorbing compounds and nanoparticles, were introduced and led to photoactivatable systems adapted to in vivo studies in small animals (zebrafish, rodent)[6]. Clinical applications relying on photodynamic therapy[7,8] or photoimmunotherapy[9] were also implemented to treat topical cancers or tumors accessible by endoscopic techniques. However, the photocontrolled systems developed until now are unable to trigger any action deeper than few millimetres in tissues because of the intrinsic low penetration of the required activating photons[10]. And this is probably today the main hurdle preventing any wide clinical application in spite of the extensive capacities of the reported systems such as drug delivery, protein and cell activity modulation, gene expression, molecular pump, slider and motor, and many more[11–15].

Herein we propose a strategy for the activation of light-sensitive systems by external stimuli without any restriction in depth. Our approach is to use high-energy waves/particles contained in ionising radiations (IRs) employed in cancer radiotherapy, such as X-ray (XR), gamma-ray (GR) or electron-beam (E) irradiations, to efficiently reach deep-tissues and then to benefit from the local conversion of carried energy into low-energy particles and species to induce specific activation of photosensitive therapeutics (Fig. 1).

Radiotherapy is a cornerstone in cancer treatment since more than half of patients will benefit from it and many efforts have been made to improve it[16]. In the first place, enhancements of IR dose were reported in the presence of iodine-based radiopaque diagnostic agent[17]. Then, nanoparticles able to increase the effect of the IR dose by radiosensitizing and/or radioenhancement effects were described, which was more recently associated to immune system activation due to the intrinsic immunogenicity of IRs[18,19]. In the last few years, several systems were reported to induce more complex and selective actions from IR stimuli[20,21]. We could separate them into two main families depending on the type of activation based on down-conversion or oxidation by ROSs. The down-converting systems, such as nanoscintillators, are designed to convert incident XR photons into UV-vis light to induce the release of cytotoxic agents such as singlet oxygen ($^1O_2$) in the case of the widely studied photodynamic therapy approaches[22]. The other systems benefit from the generated ROSs (and/or from secondary electrons)[23] to induce specific bond cleavage (mainly diselenide, disulfide, C-N, C-O, S-N, coordination bonds with metal)[24–27], DNA break[28], atom oxidation (mainly sulphur, selenium and carbon from unsaturated lipids)[29,30] leading to the disassembly of capsules, polymers or prodrugs, and the release of drug or gas (nitric oxide, carbon monoxide).

Our approach was to investigate if IR could be used in ways other than radioenhancement, bond break or atom oxidation to disassemble nanoparticle, and be employed to generate specific and non-destructive molecular actions such as molecular rearrangements described in photoactivation. For this purpose, a molecule containing both an IR-sensitive element and a photosensitive moiety was designed. High atomic number (Z) metals are known to efficiently interact with high-energy photons through their full or partial absorption and the local release of lower-energy secondary photons and electrons[16,31]. These secondary particles then induce several types of interaction with the surrounding matter as well as with the neighbouring high-Z elements leading to a ripple effect. Thus, gadolinium (Gd)-chelate was selected for both efficient interaction with IRs and detection

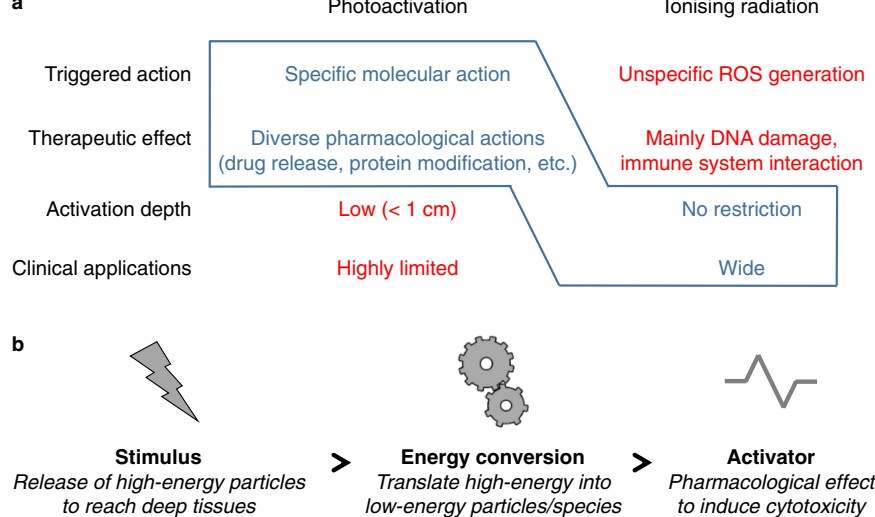

**Fig. 1 Approach to tackle deep-tissue photoactivation to overcome the current limitation preventing wide clinical applications. a** Highlight of advantages (blue) and limitations (red) of photoactivation and ionising radiation to exhibit the complementary properties of both domains which can be combined (blue delineation) for an original molecular-activation concept. **b** Our approach is based on using ionising-radiation stimulus to reach deep tissues and to benefit from local conversion of high-energy particles into low-energy particles and species suitable for molecular activation. In this way, specific pharmacological actions could be induced in biological tissues without any depth restriction.

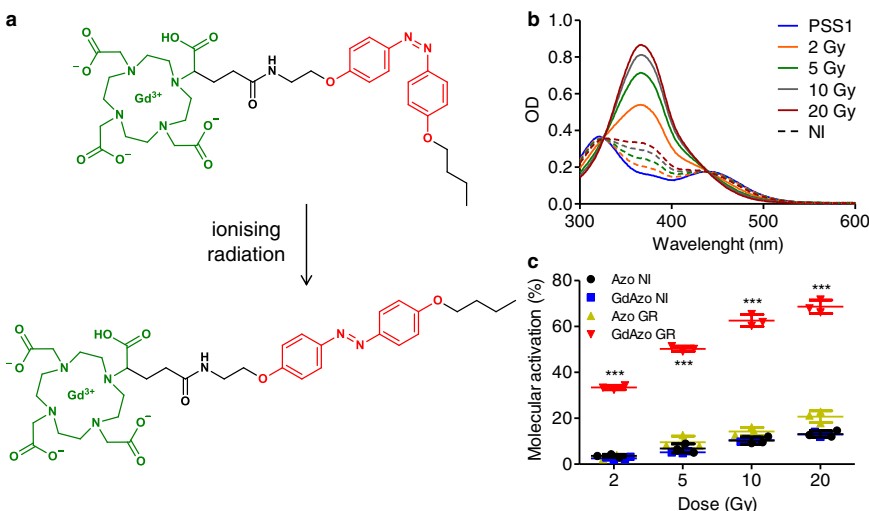

**Fig. 2 Molecular activation upon IR. a** Activation of *cis*-**GdAzo** (top) into *trans*-**GdAzo** (bottom) upon IR. **b** Absorbance spectra of *cis*-**GdAzo** compound (photostationary state 1, PSS1) recorded after GR irradiation at 2, 5, 10, and 20 Gy. The control absorbance spectra of non-irradiated (NI) *cis*-**GdAzo** compound analysed concurrently are drawn in dashed lines with corresponding color. **c** Molecular activation of *cis*-**GdAzo** and *cis*-**Azo** (control molecule without Gd atom) determined by HPLC and reported as the difference of *trans*-isomer proportion before and after GR irradiation ($n = 3$ independent experiments). The means ± standard deviations are reported. Two-way Anova (Bonferroni post-test) was used for statistical analyses (All *vs* **GdAzo** GR, $t$-values = 20.78, 30.29, 34.99 and 37.43 for **GdAzo** NI *vs* **GdAzo** GR at doses 2, 5, 10, and 20 Gy). ***$P < 0.001$. OD optical density.

by magnetic resonance imaging (MRI)[32]. On the other hand, extensive investigations proved that the azobenzene photoswitch is adapted to many pharmacological applications thanks to photocontrol of the *cis-trans* configuration switch impacting both length and polarity of the moiety[33]. Moreover, azobenzene compounds can be isomerised through different types of stimuli such as photon, heat and electron, which offers a great opportunity to implement triggering systems based on IRs.

In this study, Gd-chelate and azobenzene moiety are associated to design a special class of photosensitive system referred to as radioswitch (Fig. 2a). It is based on a photoswitch able to be activated by IRs through molecular rearrangement (isomerization) and to trigger cytotoxic effect. We investigate the underlying mechanism of the activation process and assess cancer-cell killing activity to highlight the therapeutic potential provided by this breakthrough photoswitch-activation approach.

## Results and discussion
**Design of radioswitch.** An azobenzene moiety carrying a primary amine function and alkoxyl chains at both opposite *para*-positions was prepared using standard procedures. The Gd-chelate was introduced through an anhydride opening reaction from a modified 1,4,7,10-tetraazacyclododecane-1,4,7,10-tetraacetic acid (DOTA) chelate followed by complexation with Gd. The final product **GdAzo** was synthesised over 5 steps in 33% yield (Supplementary Section 2).

Azobenzene is known for reversible shift between *trans* and *cis* configurations through isomerisation of the azo double bond after excitation by photons from UV to near-infrared energies[34]. The UV-vis absorption signatures of these two isomers significantly differ through the $\pi \rightarrow \pi^*$ electronic transition, detected at 322 and 367 nm for the *cis*-**GdAzo** and *trans*-**GdAzo** isomers respectively with much higher intensity for the latest. These spectrophotometric characteristics allow the monitoring of azobenzene isomerisation by absorbance measurement as we confirmed by proton nuclear magnetic resonance (¹H NMR) and high-performance liquid chromatography (HPLC) (Supplementary Section 3). The thermodynamically more stable *trans*-**GdAzo** isomer can be partially converted to the metastable isomer *cis*-**GdAzo** upon UV light

(photostationary state in PBS containing 90 ± 2% of the *cis*-isomer) and will then be recovered by thermal back relaxation (Supplementary Fig. 14, $t_{1/2} = 2.3$ h at 37 °C in PBS). It has to be noted that while the thermal half-life of *cis*-**GdAzo** is suitable for in vitro experiment, for any in vivo work, other systems need to be developed, with thermal half-lifes of several days to weeks.

**Molecular activation upon ionising radiation.** Primary photons from IRs have energies above the kiloelectronvolt range (1 keV ~ 1.24 nm). They have energy theoretically much too high to induce intramolecular electronic transition (such as $\pi \rightarrow \pi^*$ or $n \rightarrow \pi^*$) and exhibit the same low probability to interact with azobenzene as with any other components constituted of carbon, nitrogen or oxygen (water molecules, proteins, DNA, etc.). In contrast, IRs deliver or generate photons which have higher probability of interaction with metals of high atomic number such as Gd ($Z = 64$) through the photoelectric effect (ejection of an electron from the inner shells) since the photoelectric cross-section scales as $Z^3$-$Z^4$ depending on the incident-photon energy. This induces the release of low-energy particles and species in the very close vicinity (few nanometres) of the metal, mainly in the form of Auger electrons, lower-energy photons and reactive oxygen species (ROS)[35,36]. Monte Carlo simulations showed that such interaction induces a nanoscale dose deposition with a local increase in the equivalent energy up to several orders of magnitude by generating a large amount of low-energy species, which highlights the potential of this energy reservoir[37,38]. This increase in the radiation-dose effect has been validated for several IR sources (from kiloelectronvolt to megaelectronvolt) currently used in clinic[36,39,40].

**GdAzo** isomerisation has been assessed upon GR (662 keV, Supplementary Section 4) at irradiation doses compatible with clinical applications (2-20 Gy) (Fig. 2a). Using spectrophotometric and HPLC determinations, we discovered that *cis*-**GdAzo** was properly converted into *trans*-**GdAzo** (Fig. 2b, c and Supplementary Section 5). The activation efficacy was 33% to 69% at irradiation doses from 2 to 20 Gy (Figs. 2c, 3e). No other compound than *trans*-**GdAzo** was generated upon GR irradiation as confirmed by spectrophotometry (Fig. 2b and Supplementary

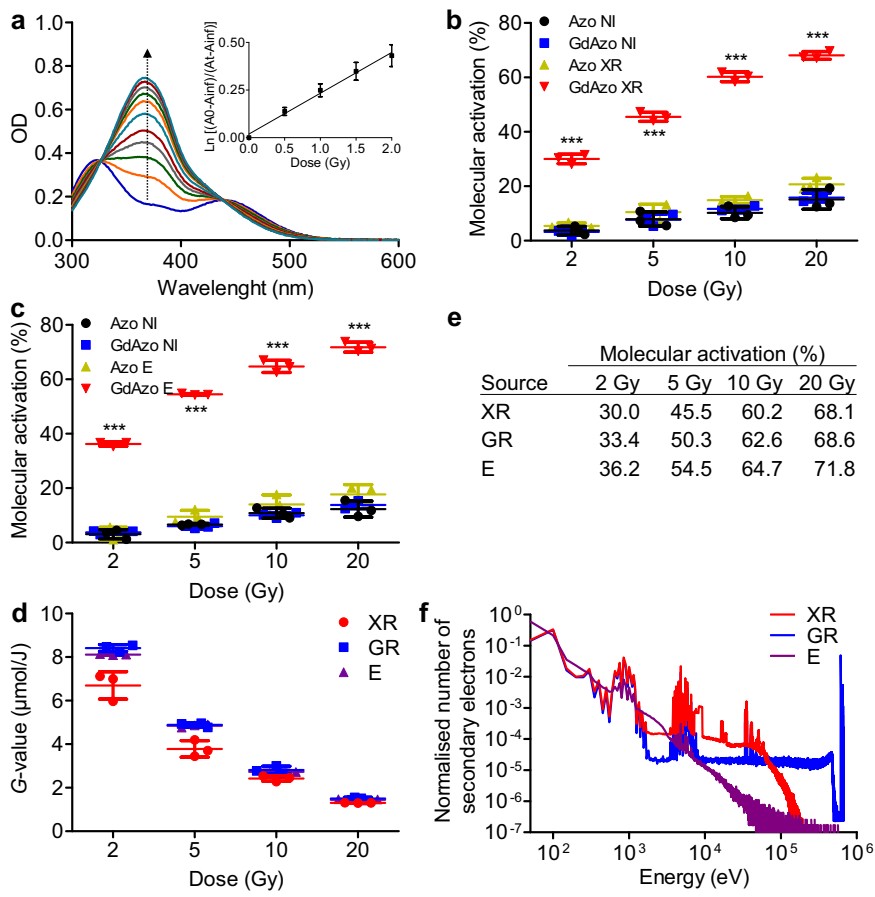

**Fig. 3 Characterisation of *cis*-GdAzo activation upon different radiation sources. a** Absorbance spectra of *cis*-**GdAzo** (PSS1 in blue) recorded after GR (0.5 Gy increment in the 0.5-2 Gy range and 1 Gy increment in the 2-8 Gy range). The graph Ln[(A0-Ainf)/(At-Ainf)] by the irradiation dose (0.5-2 Gy range) is plotted on the top-right corner, with A0, Ainf and At the absorbances of PSS1, *trans*-**GdAzo** and the mixture at a specific irradiation dose respectively. Linear regression (least-squares method, *r*$^2$: 0.9521, GraphPad Prism 5.00) were used to determine the activation constant $k = (2.1 \pm 0.2)$ *10$^{-1}$ Gy$^{-1}$ (triplicate, n = 3 independent experiments, *F*-value = 258.3, *P* < 0.0001). **b, c** Molecular activation of *cis*-**GdAzo** and *cis*-**Azo** determined by HPLC and reported as the difference of *trans*-isomer proportion before and after XR (**b**) and E (**c**) irradiations (n = 3 independent experiments). Two-way Anova (Bonferroni post-test) was used for statistical analyses (All *vs* **GdAzo** GR, *t*-values = 16.93, 23.90, 30.65, 33.01 and 19.04, 28.43, 32.09, 34.01 for **GdAzo** NI *vs* **GdAzo** GR at doses 2, 5, 10, and 20 Gy upon XR and E respectively). ***P* < 0.001. **d** Radiochemical yields (*G*-values) of *trans*-**GdAzo** determined by HPLC (corrected from thermal back relaxation) upon XR, GR and E irradiations (n = 3 independent experiments). **e** Molecular activation of *cis*-**GdAzo** upon XR, GR and E irradiations determined by HPLC. **f** Monte Carlo simulation to quantify the energy of secondary electrons emitted by the interaction of the incident primary particle on the Gd atom upon XR, GR and E irradiations. The means ± standard deviations are reported. OD optical density.

Figs. 15–17), HPLC (Supplementary Figs. 18–35) and LC-MS (Supplementary Figs. 37–44). Moreover, successive irradiations by GR and UV showed that **GdAzo** *trans-cis* isomerisation was reversible confirming that isomerisation is the molecular process happening upon IRs (Supplementary Fig. 45). The *cis*-**Azo** control compound (same structure as *cis*-**GdAzo** without Gd atom) as well as the non-irradiated *cis*-**GdAzo** and *cis*-**Azo** compounds were only slightly converted into the *trans*-isomer during the same time frame, a conversion mainly attributed to thermal back relaxation of the *cis*-isomer in the dark (Fig. 2b, c and Supplementary Section 5).

The *cis*-**GdAzo** activation upon GRs was characterised by a dose-related monoexponential increase in *trans*-**GdAzo** at low irradiation dose (< 2 Gy) and determination of a constant *k* to quantitatively characterise the activation process was then possible (Fig. 3a and Supplementary Fig. 49). The activation constants of *cis*-**GdAzo** and *cis*-**Azo** upon GRs are $k = (2.1 \pm 0.2)$ *10$^{-1}$ Gy$^{-1}$ and $k = (1.83 \pm 0.07)$*10$^{-2}$ Gy$^{-1}$ respectively (*P* < 0.0001). Another approach to quantitatively characterise this activation method is to introduce the *G*-value which is the chemical yield currently used for IR-dosimetry calculation and

consisting in the number of molecules affected per 100 eV (~1.60*10$^{-17}$ J) of energy absorbed. *cis*-**GdAzo** activation at 2 Gy was characterised by $G = 81 \pm 2$ corresponding to a molecular activation of $8.4 \pm 0.2$ μmol/J (corrected from thermal back relaxation). For comparison, hydroxyl radical (HO•) and hydrogen peroxide (H$_2$O$_2$) are generated upon water radiolysis at 0.28 μmol/J and 0.073 μmol/J yield respectively (homogeneously distributed ~10$^{-7}$ s after IR)[41]. Thus, the *cis*-**GdAzo** activation efficacy was particularly high considering the ROS released from IR sources. This type of molecular activation based on radioswitch appears very promising and open new avenues for the applications of photoswitch such as potential actuatable tools upon radiotherapy. In addition, considering the easy accessibility of radiation sources in hospitals for cancer treatment, it may turn into a very powerful technique for therapy applications.

**Mechanism of activation upon ionising radiation.** The GR photons (662 keV) used in our study mostly interact with matter through the Compton effect, which is the main physical effect occurring during clinical radiotherapy. It consists in the inelastic

scattering of the incident primary photon by a loosely-bound valence electron that is ejected. It is almost independent of the atomic number (cross-section only varies linearly with $Z$) and dominates in biological media for incident-photon energies ranging from about 50 keV to 20 MeV and 500 keV to 5 MeV for interaction with elements of $Z$ at 10 and 70 respectively[42]. To gain insight into the physical process involved during the cis-**GdAzo** activation, we investigated the impact of the incident-particle energy and its type by using an XR generator (photons of 80 keV mean energy) and a linear accelerator (LINAC, electrons (E) of 4.5 MeV) respectively (Supplementary Section 4). Indeed, 80 keV photons mainly interact with high-$Z$ elements through the photoelectric effect with an efficacy highly dependent on the atomic number of the interacting element, contrary to the Compton effect. This would suggest that cis-**GdAzo** (containing Gd, $Z = 64$) activation efficacy upon XRs could be increased compared to GR irradiation at 662 keV. On the other hand, LINAC provides high-energy charged particles mainly imparting energy through Coulomb interactions and inducing Čerenkov effect (release of UV photons), which could impact on cis-**GdAzo** activation efficacy. The obtained $G$-values and activation constants $k$ upon XR and E irradiations were in the same order as those obtained upon GR irradiation (Fig. 3b–e and Supplementary Section 5). This clearly demonstrates that the activation of cis-**GdAzo** is independent from the energy and the type of the IR, meaning that primary particle-matter interaction is not significant and that secondary particles and species emitted after this first interaction lead to cis-**GdAzo** activation.

Monte Carlo simulations (PENELOPE code)[43], allowing to determine the energy of the secondary electrons released after the interaction of the three different IR sources with Gd atoms, confirmed that the relative amount of high-energy electrons drastically differed for the three sources, whereas a similar relative amount of low-energy electrons was released from them (Fig. 3f and Supplementary Section 6). These simulations support that cis-**GdAzo** activation cannot be due to high-energy particle-Gd interaction and it is mainly triggered by the low-energy particles and species generated by energy loss from the incident primary particles, explaining the similar efficacies obtained with the different IR sources.

To go further into the understanding of the mechanism of cis-**GdAzo** activation, GR irradiations were performed in media containing scavengers able to interact with different radicals generated during water radiolysis. Results highlighted the crucial role of oxidising species in the cis-**GdAzo** activation process (Fig. 4a and Supplementary Section 7). Indeed, species which quench the hydroxyl radical HO• (tert-butanol (Fig. 4b), mannitol and ethanol) abolished cis-**GdAzo** activation except when they converted HO• into other oxidant species (sodium azide, dimethylsulfoxyde). On the contrary, electron-converting species (cadmium perchlorate) did not affect cis-**GdAzo** activation except when they converted electrons into oxidant species (sodium selenate).

Activation carried out in gas-saturated solutions validated the key role of HO• (Fig. 4c and Supplementary Section 8.1). Indeed, removing oxygen by nitrogen saturation was not affecting cis-**GdAzo** activation which bears out that hydrated electrons and hydrogen radicals (and thus the couple perhydroxyl radical/superoxide radical anion $HO_2•/O_2•^-$ resulting from their reaction with oxygen) are not involved. It has to be noted that this independence from oxygen could be very valuable for hypoxic-tumor treatment. On the other end, nitrous oxide ($N_2O$) saturation led to an increase in cis-**GdAzo** activation which unambiguously implicates the HO• as $N_2O$ converts all hydrated electrons into HO• upon irradiation of aqueous solution, resulting in doubling the production yield of HO• ($G$(HO•) = 0.56 μmol/J,

completed in ~14 ns). Furthermore, when the irradiation dose in $N_2O$ saturated solution was brought back to the generated HO• amount, the molecular activation of cis-**GdAzo** was similar as in water (Fig. 4d). For instance, 5 Gy irradiation in $N_2O$-saturated solutions (equivalent to 10 Gy in non-saturated solutions with respect to generated HO• amount, and noted "eq 10" in Fig. 4d) resulted in similar activation efficacy as 10 Gy irradiation in non-saturated solutions. The central role of HO• led us to exclude some photon-mediated interactions in the cis-**GdAzo** activation process such as the Čerenkov effect (release of UV photons from accelerated charged particles such as electrons) and the scintillation effect (release of UV photons from gadolinium atoms). Finally, comparable quenching-effects were observed upon XR, GR and E irradiations (Supplementary Section 7), which revealed a similar cis-**GdAzo**-activation mechanism using sources with different primary-particle types and energies. This confirmed that cis-**GdAzo** activation was induced by the secondary low-energy particles and species generated upon IR, which is in line with the conclusions drawn from Monte Carlo simulations.

If the hydroxyl radicals HO• was the necessary and sufficient specie for cis-**GdAzo** activation, a chemical introduction of HO• would lead to the same process. This has been investigated using Fenton chemistry which generates HO• from the dismutation of $H_2O_2$ by ferrous-iron catalyst (Fig. 5a, b and Supplementary Section 8.2)[44,45]. As expected, cis-**GdAzo** activation was induced by Fenton chemistry, while $H_2O_2$ alone was inefficient, even if some degradation was observed in these conditions (Supplementary Fig. 58).

These investigations demonstrate that cis-**GdAzo** activation is triggered by the oxidation of the azo double bond (N = N•+) by reaction with HO• generated by the low-energy particles released upon IR. Thus, the metastable cis-**GdAzo**+• radical cation is instantaneously converted into the more stable trans-**GdAzo**+• radical cation which then reduces into the final thermodynamically stable and neutral trans-**GdAzo** compound. Indeed, the isomerisation rate of cis-azobenzene-based radical cations is known to be several orders of magnitude faster than that of the corresponding neutral compound[46].

The three sources of radiation used in this study deliver low linear energy transfer radiations (LET, which is defined by the rate of energy loss per unit length of track of the particle). In low-LET radiations, the first events appear in small widely separated spurs ($10^{-16}-10^{-10}$ s time scale) and the generated radicals are homogenously spread into water at about $10^{-7}$ s after irradiation[47,48]. The HO• radiochemical yield ($G$(HO•)) is about 0.28 μmol/J (at standard dose rate); however, it can be lowered when low-energy photons (1–100 keV) are used, which is the case of the XR generator used herein[49]. Thus, $G$(HO•) delivered by both the XR generator and the GR source has been determined (Supplementary Section 9.1, Supplementary Figs. 59–63). Indirect quantification of HO• was based on HO• scavenging by coumarin[50] and quantification of 7-hydroxycoumarin (7-OH-Coum) which is the only fluorescent product released from this scavenging reaction[51]. 7-OH-Coum is proportional to HO• concentration and specific for HO• among other reactive oxygen species[52]. In the conditions used, coumarin reacts with HO• ~$10^{-7}$ s after the initial transfer of energy to water (rate constant k = $1.05*10^{10}$ L.mol$^{-1}$.s$^{-1}$)[53], which results in quantification of the homogeneously distributed HO• without interfering on the intratrack recombination of radicals occurring in the spurs. We quantified $G$(HO•) of 0.200 and 0.280 μmol/J upon XR and GR irradiations respectively (Fig. 5d when [Gd$^{3+}$] = 0), which is in line with the reported yields for similar radiation sources and dose rates[48,49]. Thus, $G$(HO•) ~$10^{-7}$ s after irradiation were 0.200, 0.280, and 0.280 μmol/J for the XR, GR and E sources respectively. Interestingly, the activation yield of cis-**GdAzo** was

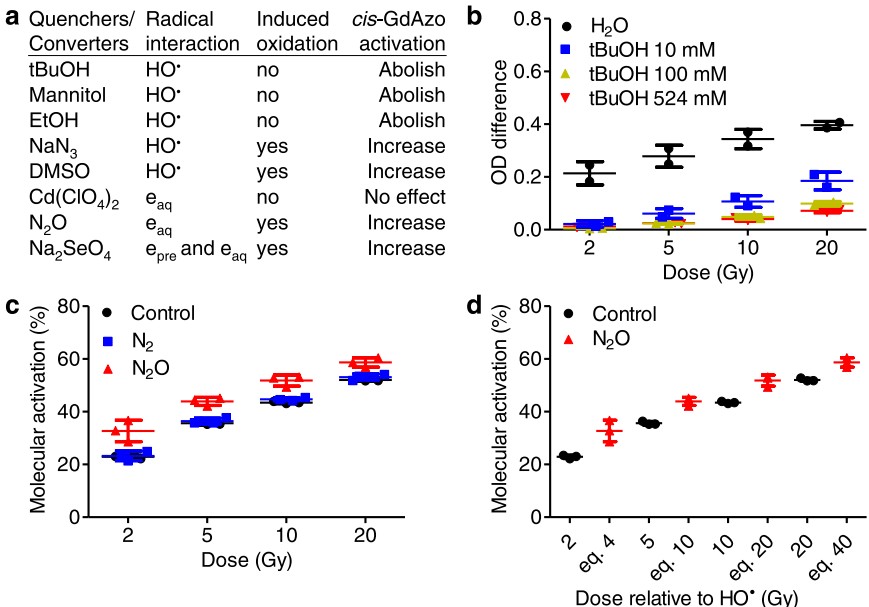

**Fig. 4 Investigation on activation mechanism using scavengers. a** Quenchers and converters used to highlight the impact of specific radicals on *cis*-**GdAzo**-activation efficacy upon XR, GR and E (cf. Supplementary Section 7 for details). **b** Absorbance difference (365 nm) of medium containing *cis*-**GdAzo** (50 μM) in the presence of tBuOH (10, 100, 524 mM (~5% v/v), control in water) before and after GR irradiation (*n* = 2 independent experiments). **c** Molecular activation of *cis*-**GdAzo** (50 μM, H₂O) upon GR irradiation in N₂ or N₂O-saturated solutions (Control without inerting) determined by HPLC and reported as the difference of *trans*-isomer proportion before and after GR irradiation (*n* = 3 independent experiments). **d** Comparison of *cis*-**GdAzo** activations in control (no inerting) and N₂O-saturated solutions in conjunction with the equivalent (eq.) amount of hydroxyl radicals HO• generated depending on doses ("eq 4", "eq 10", "eq 20" and "eq 40" relate to the doses of 2, 5, 10, and 20 Gy in N₂O saturated solutions) (*n* = 3 independent experiments). The means ± standard deviations are reported. OD optical density.

also slightly lower upon XR compared to GR and E at 2–5 Gy (Fig. 3d and Supplementary Fig. 47), even if this difference is lower than what could be expected from the $G(HO•)$ difference. This could be explained by a different kinetic regime for HO• to react with coumarin and *cis*-**GdAzo**. Indeed, the non-homogeneous energy distribution at the first stage after irradiation leads to non-homogeneous kinetics stages in the spurs.

To gain insight into the role of Gd in the activation process, $G(HO•)$ was quantified at different concentrations of Gd³⁺ ions (Fig. 5d). We observed that $G(HO•)$ was gradually increased in solutions with Gd³⁺ concentrations up to about 10–200 μM and then reached a plateau (25–500 μM) before decreasing for higher concentrations (500–2000 μM). The highest enhancement factors were 33% at 200 μM [Gd³⁺] and 20% at 500 μM [Gd³⁺] upon XR and GR respectively. It has to be noted that this study quantified the HO• that diffused into the bulk solution and not the HO• initially generated into the spurs. Thus, the saturation and decrease of $G(HO•)$ at high Gd³⁺ concentrations could be explained by the increase in the probability of HO• recombination[52]. Furthermore, addition of Gd³⁺ ions into a solution of *cis*-**Azo** led to *cis*-**Azo** activation upon GR irradiation, and the higher the Gd³⁺ concentration was, the more efficient the *cis*-**Azo** activation was (Supplementary Section 9.2, Supplementary Fig. 64). This confirmed that the presence of Gd³⁺ was required to induce activation by IR of this molecular system. Thus, the presence of Gd³⁺ in solution increased the radiochemical yield $G(HO•)$ and led to the activation of the control compound *cis*-**Azo**.

Finally, the impact of *cis*-**GdAzo** concentration on the activation efficacy was assessed to point out any cooperative effect in this process (Supplementary Section 10). The linear relation between the radiochemical yield of *trans*-**GdAzo** (*G*-value) and the initial concentration of *cis*-**GdAzo** showed there is no catalytic effect for *cis*-**GdAzo** activation, which resulted in the

same efficacy for low and high *cis*-**GdAzo** concentration (Fig. 5e). Thus, the catalytic pathway based on the activation of a substoichiometric amount of *cis*-**GdAzo** (Fig. 5c) was not involved, albeit already observed for the oxidative isomerisation of azobenzene in acetonitrile[46]. Moreover, the higher $G(trans$-**GdAzo**) at lower doses revealed a larger loss of energy at higher doses (Fig. 5e) through non-specific reactions or recombination of HO• for instance. A predictive model to estimate $G(trans$-**GdAzo**) from the initial *cis*-**GdAzo** concentration and the radiation dose was established based on a logarithmic decrease in the 2-20 Gy dose range (Fig. 5f).

**Pharmacological and cytotoxic effect.** The isomerisation of azobenzene is known to highly impact its physicochemical properties by modifying both its length and dipolar moment. Indeed, the *trans*-isomer is almost plane and has a dipole moment close to zero, whereas the *cis*-isomer exhibits an angular geometry and a much higher dipole moment[33]. Azobenzene modification has already been introduced into many systems including biomolecules, liquid crystals, or polymers to make photosensitive devices adapted to control many actions, such as complex mechanical movement, enzyme-structure modification, ion-channel opening or gene expression using UV or visible-light irradiation[5,54–60]. The *trans*-**GdAzo** has an amphiphilic structure, resulting from the hydrophilic Gd-chelate and the apolar *trans*-azobenzene moiety. We took advantage of this property to use *cis*-**GdAzo** as an IR-triggered prodrug and to demonstrate how this photochemical compound could be activated by deep-tissue penetrating lights to induce therapeutic outcomes. We, indeed, hypothesised that *trans*-**GdAzo** could act as a surfactant on cell membranes by inducing cell permeabilisation and eventually cell death. Moreover, **GdAzo** can be detected by MRI due to the presence of Gd in its structure,

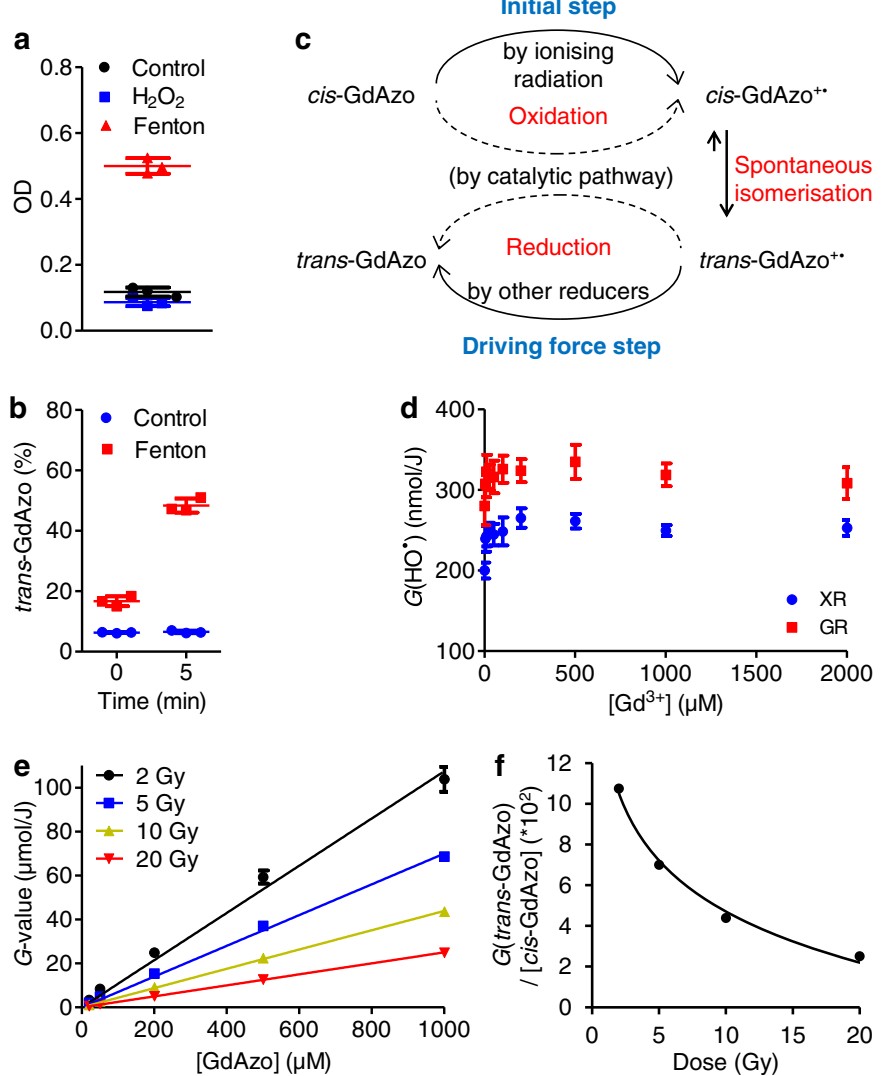

**Fig. 5 Role of HO• radicals and proposition of mechanism. a** Absorbance of *cis*-**GdAzo** (photostationary state at time 0) recorded 5 min after addition of H₂O₂ alone (bleu squares) or after running Fenton reaction (red triangle) (Control in water) (*n* = 3 independent experiments). **b** Proportion of *trans*-**GdAzo** after running Fenton reaction (Control without Fenton reagents) in *cis*-**GdAzo**-containing medium (photostationary state at time 0, just before addition of H₂O₂) determined by HPLC (*n* = 3 independent experiments). An increase in *trans*-**GdAzo** from 16.7% to 48.4% was observed 5 min after H₂O₂ introduction. **c** Proposal of a mechanism for the *cis*-**GdAzo** activation upon IRs. The initial step is the IR-induced oxidation of *cis*-**GdAzo**, leading to a radical cation on the azo double bond. The *cis*-**GdAzo**⁺• isomer then spontaneously isomerises into the *trans*-**GdAzo**⁺• which recovers its thermodynamic stable and neutral form through reduction. This last step could be induced by several reducer species in the media (hydrated electrons, hydrogen radicals, etc.) since the catalytic pathway has been rejected (*vide infra*). **d** Determination of *G*(HO•) upon XR and GR in the presence of different concentrations of Gd³⁺ by considering a conversion yield of coumarin into 7-OH-Coum of 3.1% (from ref. [52], *n* = 5 independent experiments). **e** Activation of *cis*-**GdAzo** upon GR (*G*-value) determined by HPLC (corrected from thermal back relaxation) at different initial concentration of *cis*-**GdAzo** (*n* = 3 independent experiments). **f**, Linear regression to correlate *G*(*trans*-**GdAzo**)/[*cis*-**GdAzo**] to the irradiation doses. The relation Y = −0.036*ln(X) + 0.1304 (with Y the slope of linear regressions from **e** in the unit *G*-value/µM **GdAzo** and X the dose in Gy) was obtained with *r*² = 0.9925 (Excel 2016). The means ± standard deviations are reported. OD optical density.

as experimentally confirmed by relaxivity measurements to determine MRI-detection efficacy (Supplementary Section 11.2). Such pharmacological and diagnostic properties could suit for a theranostic cancer-treatment approach.

The geometrical contrast between the *cis*-**GdAzo** and *trans*-**GdAzo** compounds was clearly revealed by calculations at the B3LYP/6-31 G* level of theory, respectively showing bended and planar structures of the azobenzene moiety (Fig. 6a). The electronic potentials at the surface correlate with the amphiphilic structure of *trans*-**GdAzo** and the theoretical dipole moment of the azobenzene moiety was decreased by 2.62 D through *cis*-*trans* conversion, which was expected to be sufficient to favour

cell-membrane disruption as compared to azobenzene-based polymers previously described[61] (Supplementary Section 12).

The amphiphilic nature and geometry of *trans*-**GdAzo** led to the self-assembly of micelles at concentration above 0.42 mM in PBS at 37 °C, as shown by fluorescence and relaxivity measurements (Supplementary Section 11). These micelles were described by small angle X-ray scattering (SAXS) as ellipsoidal aggregates (the lengths for short and long semi-axes were 37 Å and 24.5 Å respectively, Supplementary Section 13.2).

Moreover, the insertion of *trans*-**GdAzo** within model phospholipid membranes of 1,2-dipalmitoyl-sn-glycero-3-phosphocholine (DPPC) was revealed by SAXS (Supplementary

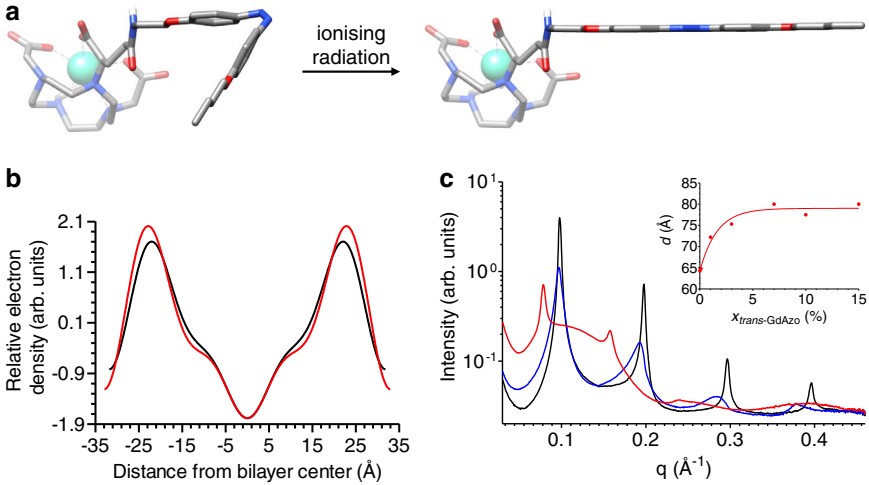

**Fig. 6 Geometrical structure of GdAzo and insertion within model phospholipid membrane. a** Structures of *cis*-**GdAzo** (left) and *trans*-**GdAzo** (right) calculated at the B3LYP/6-31 G* level of theory. **b** Electronic density of the DPPC model phospholipid membrane in the presence of *trans*-**GdAzo** (0.1%) extracted from SAXS experiments (black, red: $x_{trans\text{-}GdAzo}$ = 0, 0.1%, respectively). **c** SAXS patterns of mixtures of DPPC and *trans*-**GdAzo** (black, blue, red: $x_{trans\text{-}GdAzo}$ = 0, 0.1, 7%, respectively). Inset: Variation of the *d*-spacing as a function of *trans*-**GdAzo** molar fraction.

Section 13). Indeed, the lamellar spacing (*d*-spacing, corresponding to the sum of the bilayer and water layer thicknesses) of the DPPC lamellar phase was increased from 63.7 ± 0.3 Å to 65.9 ± 0.8 Å upon addition of 0.1 mol% of *trans*-**GdAzo**. The location of *trans*-**GdAzo** was addressed by comparing the electron density profiles of bilayers of DPPC, on one hand, and DPPC with 0.1 mol% of *trans*-**GdAzo** on the other hand (Fig. 6b). The DPPC profile reflected the electron-rich phosphatidylcholine headgroups and electron-poor hydrocarbon chains, in agreement with previous studies[62]. The distance between the phosphate groups, $d_{HH}$, deduced from the position of the two maxima was 44.0 Å in pure DPPC. The addition of *trans*-**GdAzo** slightly shifted these maxima away from the bilayer center ($d_{HH}'$ = 46.4 Å) and increased their intensity. These results are consistent with the insertion of *trans*-**GdAzo** between DPPC chains, with the electron-rich Gd-chelate moiety slightly protruding in water. Of note, the thickness of the bilayer hydrophobic core ($2D_C$ = 34.4 and 28.5 Å at 20 and 50 °C respectively)[62] enables the insertion of the hydrophobic part of *trans*-**GdAzo** whose length can be estimated at 21.8–22.6 Å, according to the molecular model (cf. Supplementary Section 12). The addition of increasing amounts of *trans*-**GdAzo** to DPPC model membrane led to a progressive shift towards smaller scattering vectors *q* of the Bragg peaks ($q = 4\pi \, sin(\theta)/\lambda$ where $2\theta$ the scattering angle and $\lambda$ the radiation wavelength; Fig. 6c). At room temperature, the *d*-spacing increased from *d* = 64.3 Å in DPPC to *d* = 80 Å in DPPC containing 7% of *trans*-**GdAzo** and then remained constant up to 15% of *trans*-**GdAzo**. This change in *d*-spacing was accompanied by a broadening and a decrease in intensity of the Bragg peaks, indicating a more disordered lamellar phase involving fewer bilayers. A broad maximum centred on $q \approx 0.12$ Å$^{-1}$, compatible with scattering from micelles, was also clearly observed for *trans*-**GdAzo** molar fraction ≥3%. These results suggest the coexistence of stacked bilayers and micelles, both of which would be mixed structures containing *trans*-**GdAzo** molecules. The interaction of *trans*-**GdAzo** with DPPC bilayers was further supported by differential scanning calorimetry (DSC) (Supplementary Section 13). The model emerging from all the SAXS and DSC results is that the partial solubilisation of phospholipid bilayers by *trans*-**GdAzo** at high concentration involved the formation of mixed micelles consisting of phospholipid nanodiscs whose hydrophobic edges were shielded from water by *trans*-**GdAzo** molecules.

The cell permeabilisation in the presence of either *cis*-**GdAzo** or *trans*-**GdAzo** was assessed by microscopic examination of cancer cells (PANC-1) incubated with propidium iodide (PI). While neither cell permeabilisation, nor cytotoxicity were observed with *cis*-**GdAzo**, the active *trans*-**GdAzo** isomer induced cell permeabilisation in few minutes after incubation (Fig. 7a, b and Supplementary Figs. 78–85). The cell permeabilisation of *cis*-**GdAzo** after activation into *trans*-**GdAzo** upon GR (2 Gy) was significantly higher compared to (i) exposure to *cis*-**GdAzo** without GR or (ii) GR in the absence of *cis*-**GdAzo** (Fig. 7c–e and Supplementary Figs. 86, 87). These observations revealed that *trans*-**GdAzo** could induce a loss of integrity for some cells leading to collapse (Fig. 7l and Supplementary Videos 1–3), which was attributed to the partial solubilisation of the phospholipid bilayers as shown by SAXS on model membrane (Supplementary Section 13). A similar collapse of cell structure due to the rapid breakdown of cell membrane has previously been reported for high concentration of the surfactant Triton X-100[63].

Electron energy-loss spectroscopy coupled to transmission electron microscopy (EELS-TEM) confirmed the presence of Gd in the permeabilised cells and revealed a heterogeneous distribution in the cell cytoplasm. The *trans*-**GdAzo** compound accumulated in specific cytoplasmic areas and was not homogenously diffused (Fig. 7i and Supplementary Section 14.5). At this stage, we were wondering if *cis*-**GdAzo** activation could induce a lethal action by itself or favour drug penetration inside resistant cancer cells. The cytotoxicity induced by *cis*-**GdAzo** activation was assessed on a gemcitabine (Gem)-resistant cancer cell line (CCRF-CEM ARAC 8 C, human T lymphocytes) which does not express the hENT-1 membrane receptor required for the Gem to penetrate inside the cell, before phosphorylation and inhibition of DNA synthesis[64]. First, the permeabilization of the Gem-resistant cell membranes by *cis*-**GdAzo** upon GR (2 Gy) was confirmed using PI and flow cytometry quantification (Fig. 7f–h and Supplementary Section 14.4). To assess cytotoxicity, the cells were treated for 1 h with *cis*-**GdAzo** upon irradiation by GR (2 Gy) and the cell viability was assessed after four-day incubation in standard conditions. The treatment performed in the presence or absence of Gem confirmed the killing activity of *cis*-**GdAzo** upon IR and showed that the additional cytotoxicity effect of Gem was not significant (Fig. 7j, k and Supplementary Figs. 90, 91). The absence of any increase in Gem intracellular-accumulation in the resistant

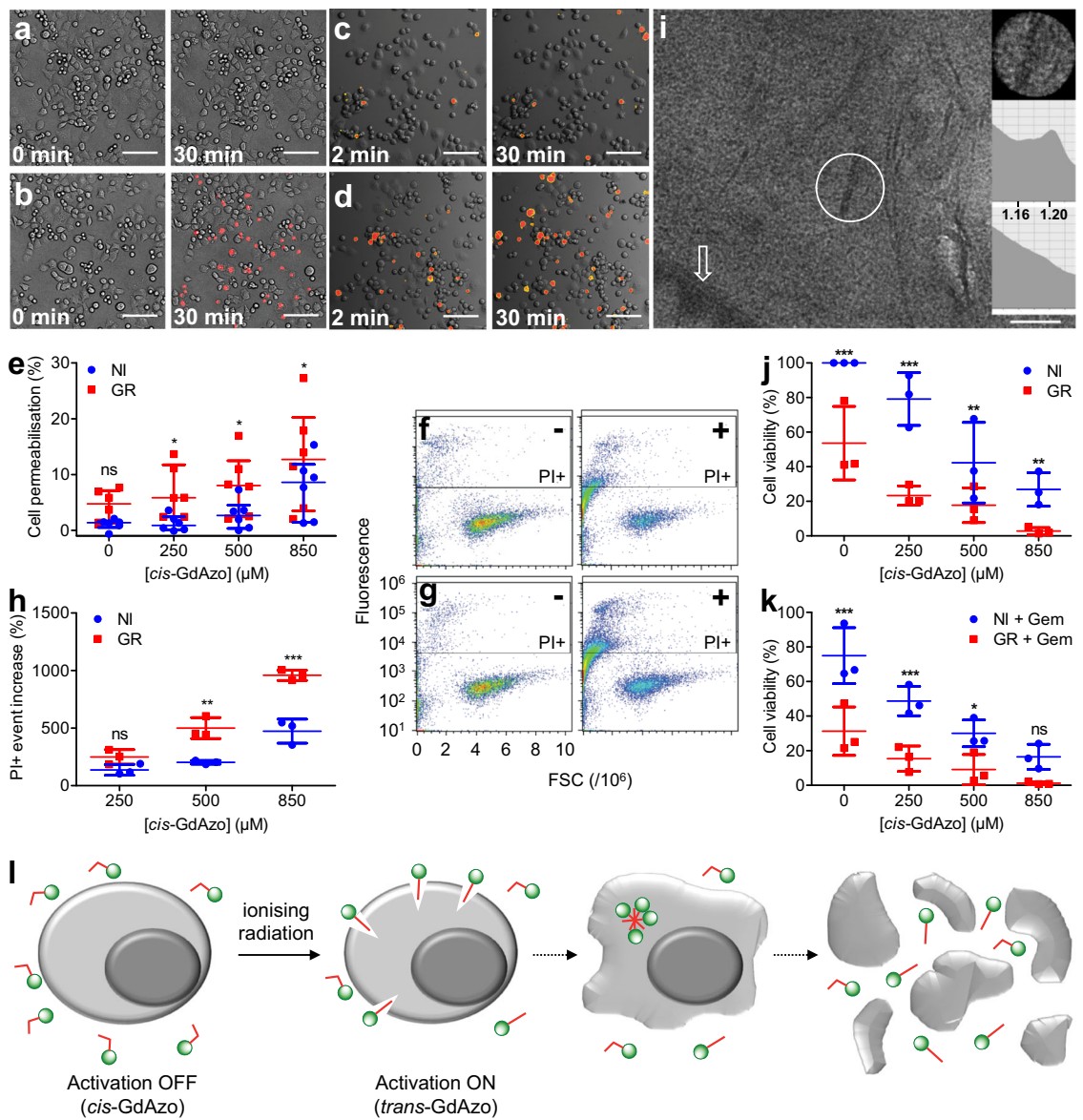

**Fig. 7 Cell cytotoxicity triggered by *cis*-GdAzo activation upon IR. a, b** Confocal microscopy of cancer cell (PANC-1) in the presence of PI before (0 min) and 30 min after incubation with either inactive *cis*-**GdAzo** (**a**) or active *trans*-**GdAzo** (**b**) (4 repetitions, permeabilised cells in red). **c, d** Confocal microscopy of cancer cell (PANC-1) in the presence of PI at 2 min and 30 min after *cis*-**GdAzo** introduction followed by GR (**d**) or no (**c**) irradiation (permeabilised cells in red). **e** Quantification of cell permeabilisation (PANC-1) by confocal microscopy after treatment with *cis*-**GdAzo** upon GR (*n* = 6 biologically independent samples, processed by a custom script, *t*-values = 3.110, 2.941, 2.815 at 250, 500, 850 µM). The medians ± interquartile ranges are reported. **f, g,** Flow cytometry of cancer cells (CCRF-CEM-ARAC-8C) in the presence of PI and *cis*-**GdAzo** (absence (-) or 500 µM (+)) upon GR (**g**) or without (**f**) irradiation at 30 min post-treatment. **h** Quantification of cell permeabilisation (CCRF-CEM-ARAC-8C) by flow cytometry after treatment with *cis*-**GdAzo** upon GR at 30 min post-irradiation (*n* = 3 biologically independent samples, *t*-values = 5.254, 8.565 at 500, 850 µM). The relative increase in PI-positive events compared to medium without *cis*-**GdAzo** is represented. **i,** EELS-TEM on cancer cell (PANC-1) in the presence of *trans*-**GdAzo**. Gd was detected in localised cytoplasmic areas (circle, top EELS spectrum, keV axis unit) and not in the surrounding cytoplasm (bottom EELS spectrum), cell membrane or nucleus (arrow). **j, k** Cell viability of Gem-resistant cancer cells (CCRF-CEM-ARAC 8 C) 4 days after treatment with *cis*-**GdAzo** upon GR in the absence (**j**) or presence (**k**) of Gem (*n* = 3 biologically independent samples, *t*-values = 6.854, 8.258, 3.636, 3.557 at 0, 250, 500, 850 µM in the absence of Gem and 6.454, 4.927, 3.099 at 0, 250, 500 µM in the presence of Gem). **l** Representation of the impact of *cis*-**GdAzo** on cancer cells upon IR as assumed from the microscopy experiments. The means ± standard deviations are reported unless otherwise specified. Two-way Anova (Bonferroni post-test) was used for statistical analyses (NI *vs* GR). ns: not significant, *$P < 0.05$, **$P < 0.01$, ***$P < 0.001$. Scale bars = 125 µm.

cancer cells in the presence of *trans*-**GdAzo** was confirmed by using radiolabelled Gem (Supplementary Fig. 93) and was attributed to dead-cell collapse. Even if the synergistic cytotoxic effect of *cis*-**GdAzo** and IR will need to be improved, this approach brought out that nearly complete cancer-cell death can be achieved at high concentration of *cis*-**GdAzo** upon IR (remaining of 2.8% living cells). It has to be noted that no

cytotoxicity was detected using a control compound containing the Gd-chelate without azobenzene modification (Dotarem®), thus confirming the need of the presence of the azobenzene moiety and refuting the idea that cytotoxicity could be due to the Gd-induced radiosensitisation only (Supplementary Fig. 92).

Photosensitive systems have been developed for many decades but their translation into clinic reveals to be very disappointing

mainly due to the non-penetrating lights they need for activation. We overcome here this limitation by designing a radioswitch, i.e. a photoswitch system adapted to IR at low clinical dose such as currently used for radiotherapy, and inducing a cell-permeabilising and destructuring effect. Thanks to the Gd-chelate moiety associated to the photosensitive system, theranostic approach should be possible by in vivo MRI detection of the prodrug before triggering a localised therapeutic action. Taking advantage of the various pharmacological actions able to be triggered by photoswitch systems, many cellular manipulations and therapeutic approaches could come out as the cell permeabilisation described in this work. This exciting new development in photochemistry opens the way towards novel opportunities in the translation of photoswitching molecular tools currently limited to research area into clinical applications.

## Methods

The synthesis procedures and characterisation of the **Azo** and **GdAzo** compounds are described in the Supplementary Sections 1–3. The methods to assess activation upon IR are described in the Supplementary Sections 4, 5. The Monte Carlo simulation is reported in the Supplementary Section 6. The study on activation mechanism upon IR in the presence of various quenchers/converters is detailed in the Supplementary Section 7 Investigations on the role of hydroxyl radicals (experiments under gas saturation and Fenton chemistry), Gd (quantification of hydroxyl radicals and activation of *cis*-**Azo** in the presence of Gd) and the impact of *cis*-**GdAzo** concentration are reported in the Supplementary Sections 8–10. The physicochemical characterisation, DFT calculations and study of **GdAzo** aggregation (SAXS) and interactions with cell membrane model (SAXS and DSC) are described in the Supplementary Sections 11–13. The methods to assess in vitro cytotoxicity are reported in the Supplementary Section 14. The main methods are listed below even if more details are provided in the Supplementary Sections.

**Quantification of *cis*-GdAzo activation upon IR**. *trans*-**GdAzo** and the control compound without Gd (*trans*-**Azo**) (50 μM, 200 μL, PBS) were introduced in two 96-well microplates. Both microplates were irradiated by UV (365 nm, $0.817 \, \text{mW.cm}^{-2}$, 5 min) to obtain the PSS1 containing a majority of the *cis*-isomer (90 ± 3%). One microplate (plate 1) was kept in the dark and used as control (non-irradiated by IR) whereas the second microplate (plate 2) was irradiated upon incremental doses of IRs (2, 3, 5 and 10 Gy). After each irradiation, absorbance and HPLC (method C) analyses were performed on the non-irradiated (plate 1) and IR-irradiated (plate 2) compounds. A time delay between the UV irradiation of plate 1 and plate 2 was introduced to analyse both the control (plate 1) and the IR-treated (plate 2) compounds concurrently. The relative amount of each isomer was obtained by running HPLC and the molecular activation (%) was determined by the difference in proportion of the *trans*-isomer in the media. The experiment was repeated 3 times independently.

For investigation on the activation mechanism using scavengers, the *cis*-**GdAzo** compound was dissolved in different media: water (control), aqueous solutions of *tert*-butanol (10, 100, 524 mM), mannitol (10, 100, 524 mM), ethanol (10, 100, 524 mM), sodium azide (1, 10 and 50 mM), dimethylsulfoxyde (15% v/v), sodium selenate (1 and 25 mM) and cadmium perchlorate (20 mM). After each irradiation, absorbance analyses were performed on the non-irradiated (plate 1) and IR-irradiated (plate 2) compounds (triplicate).

***cis*-GdAzo activation under N₂ and N₂O gas saturation**. *trans*-**GdAzo** (50 μM, 200 μL, H₂O) was introduced in a 96-well microplate before UV irradiation (365 nm, $0.817 \, \text{mW.cm}^{-2}$, 10 min, *cis*-isomer at 90 ± 3%). The compound was then introduced into two sealed glass tubes (700 μL) and the medium was saturated by N₂ or N₂O gas bubbling for 15 min. One of the glass tube was kept in the dark (control) whereas the second one was irradiated upon incremental doses of GRs (2, 3, 5 and 10 Gy). After each irradiation, HPLC analyses were performed (method C) and the molecular activation (%) was determined by the difference in proportion of the *trans*-isomer in the media. The experiment was repeated 3 times independently.

***cis*-GdAzo activation using Fenton chemistry**. *cis*-**GdAzo** (1.000 mL, final concentration 50 μM, H₂O), EDTA (9.00 μL, final concentration 75 μM, H₂O) and FeCl₂ (9.00 μL, final concentration 75 μM, H₂O) were introduced in an 1.5 mL Eppedorf tube. A first HPLC analysis was run before the introduction of H₂O₂. The Fenton reaction was started by the addition of H₂O₂ (4.41 μL, final concentration 50 mM, H₂O) into the mixture which was stirred (orbital) in the dark. The proportion of *trans*-**GdAzo** was determined by HPLC (method C) 5 min after the addition of H₂O₂. The experiment was repeated 3 times independently.

**Quantification of hydroxyl radicals**. Coumarin (600 μL, final concentration 0.5 mM, H₂O) and GdCl₃ (0-168 μL, final concentrations from 0 to 2000 μM, H₂O) were mixed and introduced in a 96-well microplate (200 μL, H₂O, duplicate). The

microplate was irradiated upon incremental doses of ionising radiations (final doses of 0, 2, 5, 10, 15, 20, 25, 30 and 40 Gy). After each irradiation, fluorescence (ex: 355 nm, em: 460 nm) was measured. The experiment was repeated 5 times independently. The standard curve of 7-OH-Coum led to quantify the production of 7-OH-Coum by increasing the radiation dose at different $Gd^{3+}$ concentrations. The slope of these lines (related to $G$(7-OH-Coum)) were plotted against the $Gd^{3+}$ concentrations to assess the impact of $Gd^{3+}$ on HO• production. Finally, $G$(HO•) was determined by applying a conversion yield of Coum into 7-OH-Coum of 3.1% (cf. Supplementary Section 9).

**Cell permeabilisation of *cis*-GdAzo upon GR using confocal microscopy**. The cells (PANC-1) were seeded (10,000 cells in 200 μL/well) in imaging chamber and maintained in culture medium in a humid atmosphere at 37 °C with 5% CO₂. 24 h postseeding, culture medium was replaced by PBS (95 μL) and propidium iodide (PI, 5 μL, final concentration 1 μM) was added in the medium. 15 min after addition, *cis*-**GdAzo** (100 μL, final concentration 0, 250, 500, or 850 μM, PBS) was introduced in the medium. A first set of images was acquired at this stage and then the imaging chamber was irradiated (GR, 2 Gy). The imaging chamber was kept in the dark at 37 °C using a stage heater and images were acquired each 5 min for 30 min. A similar procedure was used for the non-irradiated control experiment except that the imaging chamber was not irradiated upon GR. The cells were observed with an inverted Nikon microscope (10x dry objective lens). The red fluorescence emission of PI (ex: 561 nm, em: 598-672 nm) and transmission images were collected. Four images per well were acquired and two wells were used for each concentration of *cis*-**GdAzo**. The experiment was repeated 3 times independently and each well was considered as a biological independent replicate. The number of cells on images acquired before irradiation was manually counted and the number of fluorescent cells on images acquired before and after GR irradiation was automatically determined using an in-house script for ImageJ software[65] (Version 1.50i completed with Adjustable Watershed plugin).

**Cell permeabilisation of *cis*-GdAzo upon GR using flow cytometry**. Just before the experiment, cells (CCRF-CEM ARAC-8C) were dispersed in PBS and transferred in 96-well microplate (40,000 cells in 90 μL/well). *cis*-**GdAzo** (90 μL, final concentration 0, 250, 500, or 850 μM, PBS) was introduced before irradiation of the medium by GR (2 Gy). Then, PI (5 μL, final concentration 1 μM) was added and the medium was maintained in the dark at room temperature. Flow cytometry analyses were run at 15, 30, and 45 min after irradiation using a BD Accuri™ C6 Plus flow cytometer (runs of 30 μL, 100 μL/min, no threshold). A similar procedure was used for the non-irradiated control experiment except that the microplate was not irradiated upon GR. Cell membrane permeabilisation was quantified by numbering the PI-positive events (duplicate) and represented as the relative increase in PI-positive events compared to medium without *cis*-**GdAzo**. The experiment was repeated 3 times independently. The data were treated using the BD Accuri C6 Plus software (version 1.0.27.1) and the representative images were obtained using the Flowjo software (version 10.7.1).

**Cytotoxicity of *cis*-GdAzo upon GR**. Just before the experiment, cells (CCRF-CEM ARAC-8C) were dispersed in PBS and transferred in 48-well microplate (40,000 cells in 80 μL/well). Gemcitabine (Gem, 20 μL, final concentration 0.1 μM) or PBS (20 μL) and *cis*-**GdAzo** (100 μL, final concentration 0, 250, 500 or 850 μM, PBS) were added before irradiation of the medium by GR (2 Gy). The medium was maintained in the dark in a humid atmosphere at 37 °C with 5% CO₂ for 1 h. Then, culture medium (600 μL) was added in the wells and the cells were washed by 3 centrifugation cycles (300 g, 5 min). Cells were dispersed in culture medium (600 μL) containing Gem (0 or 0.1 μM final concentrations) and were maintained in a humid atmosphere at 37 °C with 5% CO₂ for 4 days. Living cell number was then determined by cell counting in presence of trypan blue 1:1 v/v (triplicate). The experiment was repeated 3 times independently. The cell viability was expressed as the ratio of living cell number after treatment to living cell number without any treatment (non-irradiated, no Gem and no *cis*-**GdAzo**).

**Reporting summary**. Further information on research design is available in the Nature Research Reporting Summary linked to this article.

## Data availability

All characterisation data and experimental protocols to evaluate the conclusions in the paper are available in the manuscript and/or the Supplementary Information. Moreover, source data is available for all the figures and Supplementary Figures in the associated Source Data file, and the main data generated in this study have been deposited in the Zenodo database[66] under accession code https://doi.org/10.5281/zenodo.6379759.

## Code availability

The custom codes developed for this study are available at the Zenodo database[66] under accession code https://doi.org/10.5281/zenodo.6379759. They can be run with imageJ

(version 1.50i completed with Adjustable Watershed plugin, to determine cell permeabilisation from the optical microscopy data) and with UCSF Chimera (version 1.14, to classify and display polar and non-polar surfaces from the DFT calculations).

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

## Acknowledgements

We would like to acknowledge financial support from the French National Research Agency (ANR PDOC20130008-01 grant), from the Université Paris-Saclay for the "Initiative de Recherche Stratégique" IRS NanoTheRad project (contract No. 160573) and from ITMO Cancer of Aviesan on funds administered by Inserm. I. M.-R. acknowledges the financial support from the Spanish Ministry of Science, Innovation and Universities (Ramon y Cajal research fellowship RYC2018-024043-I). The authors wish to thank Claire Lovo and Dr Marie-Noëlle Soler from the PICT-IBiSA Orsay Imaging facility at Institut Curie, Charlène Lasgi from the Cytometry platform of CurieCoreTech at Institut Curie, as well as Dr Valérie Nicolas from the UMS IPSIT of Université Paris-Saclay (US31 INSERM, UMS3679 CNRS, Plateforme d'imagerie cellulaire, MIPSIT) for providing access to the chemical imaging and flow cytometer facilities and for their help in data treatment. Dr Javier Perez (Synchrotron SOLEIL, Saint-Aubin, Gif-sur-Yvette, France) is warmly acknowledged for his help with SAXS experiments. The UMR9187 / U1196 (Institut Curie, Orsay, France) is gratefully acknowledged for giving access to analytical instruments and the Université Paris-Saclay IT department for providing computing resources, as well as Dr Bertrand Fournier for his help setting these resources up. Dr Vincent Favaudon, Dr Sophie Heinrich and Dr Matteo Martini are warmly acknowledged for the fruitful discussions. We also would like to acknowledge the NMR facility from the Université Claude Bernard Lyon 1 (Centre Commun de RMN) for advices.

## Author contributions

G.Bo. devised, designed and supervised the project. G.Bo., C.L.M., F.A., and A.G.-V. carried out organic syntheses. G.Bo. and S.Mé. conducted the relaxivity measurements. G.Bo. and A.G.-V. performed the experiments to determine conversion upon IR. G.Bo., A.G.-V. and F.P. carried out and discussed the experiments upon IR in the presence of scavengers. G.Bo. performed the experiments to validate the role of hydroxyl radicals (using scavengers, gas-saturated solutions, Fenton chemistry) and to investigate the mechanism of activation (using different concentrations of GdAzo, quantification of hydroxyl radicals). I.M.-R. performed the Monte Carlo simulation. G.Be. and G.Bo. conducted the DFT calculations. C.B., F.-X.L., and G.Bo. performed the SAXS and DSC experiments. S.Ma., S.T., and G.Bo. carried out the EELS-TEM experiment. G.Bo. carried out the experiments on cancer cells using microscopy and flow cytometry. G.Bo., C.R., and F.P. carried out the cytotoxicity experiments on cancer cells. G.Bo., P.C., and S.Mu. discussed the results. G.Bo. wrote the article. All the authors discussed the results and validated the last version of the article.

## Competing interests

A patent associated with this work has been filed by G.Bo., P.C., S.Mu. and F.P.(WO/2020/161308). All other authors declare no competing interests.
