## [Peer Review File · Nature Communications]

Breaking photoswitch activation depth limit using ionising radiation stimuli adapted to clinical applicationREVIEWER COMMENTS

Reviewer #1 (Remarks to the Author):

The manuscript by Bort et al. deals with an interesting system, which however from the results presented in the paper only gives a small cytotoxicity benefit if any and under not entirely clarified conditions. The claims of the authors are quite extravagant with respect to what they present and should be toned down, whereas there is no clear mechanistic explanation on how the system works despite the efforts of the authors, and at the end a hypothesis is put forward which cannot account for the nature of the findings.

So my recommendation is rejection at this stage at which the results are very premature for publication, with the possibility to resubmit in the future when the results are more coherent and substantial, and a more plausible mechanism is proposed.

More specifically:

I would recommend a full language restructuring both from the gramatic/syntax point of view but also regarding the scientific terms. As an example "azote" is perfectly fine in French or Greek but is not used in English. Also the authors' favourite expression "lower quanta of energy" leaves many questionmarks regarding its meaning in most of the places used.

Main points:

1. The title needs to be significantly toned down, as the authors suggest that they are the ones to break the photoactivation limit and with a clinical application. A brief review of the recent bibliography will show them that this limit has been broken before, whereas they must explain how exactly they adapted their work to clinical application. This applies to the whole of the text which should be restructured and toned down to match the findings; for example the introduction does not refer at all to prior attempts to activate photoactive compounds directly or indirectly by ionising radiation!!

2. In Fig. 2 we see a significant difference in the non-irradiated Azo or Gd Azo between 2 and 20 Gy . How do the authors explain that? This is a recurring theme also in Fig. 3 a and b... The non-irradiated Azo or GdAzo should be the same for 2 or 20 Gy, simply because this sample is not irradiated (0 Gy)!! What accounts for these sizable differences which are not covered by the error bars.

3. In line 213 the authors based on their results they claim that the activation of cis-GdAzo is independent of the type of radiation and is the same for photons and electrons. How can they account for this? They try to produce a mechanism, but this does not agree with the results and is not fully substantiated. They attribute their system's action to HO· and associated oxidation, but: did they show that the same amounts of HO· are produced in all the cases of ionizing radiation? Why don't they use the well known HO· quencher mannitol to see the effect? The authors refer to the HO· conversion to H₂O₂, but how about the action of H₂O₂ on the cisGdAzo, have they checked it? And what about the reverse? How would the conversion of H₂O₂ to HO· affect the cisGdAzo? In other words, would the Fenton reaction which takes place within the cells, lead to spontaneous activation of cisGdAzo?

And what happens to the cisGdazo activation in conditions of hypoxia? The fact that XR, GR and e- have the same activation effects remains a mystery.

The authors have a long road ahead to understanding and thus harnessing their system.

4. The authors need to identify how deeply into the membrane their molecule digs, since the hydrophilic head pulls the molecule out of the membrane. I do not agree with the authors extrapolations from Fig. 4. Firstly, one glance at the big error bars is enough to see that in Fig. 4g there is no statistical significance (difference) between the white(NI) and black (GR) bars. From 4h we can see that the compound confers a significant toxicity on its own (70%) at 850uM and if we consider the negative error bar of 0 concentration for the black bar (irradiated)then we see that

irradiation could cause a ~70% toxicity, which makes the 250 uM and 500 uM results non significant (the 500uM one was non-significant anyway because of the huge white bar error). That leaves the case of the 850uM where the compound chemical toxicity is already 70%. Similar extrapolations can also be made for the case of gem. Also about gem the authors state that the amount of gem does not increase in gem-resistant cells in the presence of trans-GdAzo. But if trans-GdAzo indeed permeabilizes the membranes (as the authors claim) to the point that PI can flood in, then why does not gem go in through the permeabilized membranes?

5. PI is not very reliable under prolonged incubation as it might stain the cells even without cell permeabilization. Figs. 4BC and 4DE are shown in different magnification modes. It is difficult to compare the extent of PI signal between the groups. Those panels of the same magnification are necessary to provide. The same situation in supplementary Sections 11.2 and 11.3.

6. it seems that the GdAzo-caused enhancement in cell killing was quite low without even taking the error bars into account. If one applies simple mathematics to the data presented in Fig. 4h and crude numbers taken from the presented figure in then only 10% enhancement can be obtained: $(NIGdAzo - GRGdAzo) - (Ctrl - GR) = 80 - 25 - (100 - 55) = 55 - 45 = 10\%$
This is too low a synergy also presented in the showcase results. Perhaps it would be a good idea for the authors to present their results in different experiments so the median gain from the use of cisGdAzo can become clearer.

7. How does the molecule (Gd-Azo) exist more stably in its natural form? Is it in the cis form, or in the trans form? The understanding is that the cis form does not kill but the trans one kills. If the molecule originally exists in the the cis form then disregard the question. However if the compound naturally exists more stably in the trans form (as stated in the manuscript text) and the authors needed to switch it to cis by UV light, before giving it to cells and applying the radiation to switch it again to trans then this raises many questions! That is because in any case the cis compound will thermodynamically convert to trans within 12-24h (according to the authors) and will kill the cells anyway! perhaps this is not a disaster in the case of cell cultures (in vitro) where one can remove the compound before the thermodynamic restructuring, but it would be a disaster in the case of the patient, where if the compound is not fully metabolised and excreted in 12-24h then there will be a big spontaneous toxicity problem. The authors need to clarify their use of the compound throughout the manuscript text, and make it easier to comprehend for the reader as it currently is a bit obscure.

8. Lines 113-115. The authors supposedly refer to other Azo derivatives absorbing in near-infrared region as Fig. 2B shows absorption up to about 500nm. That must be clarified because excitation wavelength in the setup of confocal microscopy falls into that region.

minor comment

Line 74-75: The authors refer to all ionizing radiation as particles and while this is more correct for electrons or protons it is not entirely correct for photons.

Reviewer #2 (Remarks to the Author):

This is a great paper. It makes a very substantial new contribution to the application of photochemistry to biomedicine. It is very thorough and represents a substantial amount of work. The central idea is that ionizing radiation, which can penetrate deep into tissues, can lead to the production of an oxidant (e.g. an OH radical) by a Gd chelate. If this occurs close to a cis-azobenzene (e.g. covalently linked to the gd chelate), it can trigger thermal cis to trans isomerization of the azobenzene through a $N=N^+$ azo radical. This builds substantially on conceptual work by Hecht (as referenced), and convincingly demonstrates that cis-to-trans azo

isomerization be done with ionization radiation in practice. I am convinced by the extensive data on the compounds analyzed in solution. To my mind, this is enough of a conceptual, and practical advance to warrant publication in Nature. The authors take the work a bit farther, however, and analyze cell permeabilization by the azo-linked Gd chelate, likely via a membrane disruption mechanism. While this data does seem to show an effect, I would be more convinced if flow cytometry data were used to quantitatively measure degree of permeabilization, rather than imaging data. In any case, to my mind, this is likely not the compound that would actually be employed in a clinical setting. Instead chemists and photomedicine experts can begin to design ideal compounds now that the photoactivation depth limit has been broken! One thing that would make the work even more compelling is if the authors could inject the cis compound into a mouse and recover the trans after irradiation! But perhaps this is too difficult if the thermal half life is 2-3h anyway (at room temp?). Congratulations!
Andrew Woolley

Reviewer #3 (Remarks to the Author):

The paper 'Breaking photoactivation depth limit using ionising radiation stimuli adapted to clinical application' by Bort et al. describes the application of ionizing radiation to induce isomerization of azobenzene conjugated to Gd-DOTA complex. The paper is clearly very intriguing and as such it should be of interest to Nature Communications. However, I think that the paper should be accepted only when the mechanism behind this isomerisation is properly proven, which is not the case yet. I have the following comments/suggestions.

- The authors explain that high Z materials have better adsorption of high energy radiation, such as gamma's and X-rays. This is certainly true but the papers the authors refer to provide experiments using nano-particles made of high Z materials while they use Gd-DOTA complexes, hence much less Gd ions are probably present. The interaction probability in this case will depend on the concentration of Gd, something that is entirely not discussed and also not mentioned in the simulation part.
- I am also not entirely convinced by the use of scavengers. Why use NaN₃, this is a typical scavenger for Singlet Oxygen? Many of the used scavengers may influence also the production/elimination rate of other species. Butanol for instance also affects hydrogen peroxide presence. The scavenging experiments are actually much more complex than presented in the paper.
- The most intriguing part is the lack of influence of radiation type. The same effects are observed for gamma's, X-rays and electrons. I miss here the G values for OH radical for each radiation source. The experiments suggest that the same amount of OH radicals are produced. Is that true?
- The presence of Gd suggests to lead to higher isomerisation rate. If secondary electrons leading to OH radicals were the driving force then I would expect that it is not necessary to have Gd linked to the azobenzene. A Gd solution if well mixed should work similarly. Why was that not tested by the authors? An experiment in which the Gd concentration is varied would shine more light on the possible role that this element plays.
- Furthermore can the authors determine how much more OH radicals are formed in the presence of Gd in comparison to samples without Gd? Why was that not attempted?
- To check the effect of OH radicals a chemical induction of OH radicals such as Fenton reactions can be applied. Why didn't the authors try that?
- Finally, what about possible scintillation effects of Gd? I do not consider it very likely but I think that this should be discussed.

Response to reviewers' comments - NCOMMS-20-22126-A**Reviewer #1 (Remarks to the Author):**

The manuscript by Bort et al. deals with an interesting system, which however from the results presented in the paper only gives a small cytotoxicity benefit if any and under not entirely clarified conditions. The claims of the authors are quite extravagant with respect to what they present and should be toned down, whereas there is no clear mechanistic explanation on how the system works despite the efforts of the authors, and at the end a hypothesis is put forward which cannot account for the nature of the findings.

So my recommendation is rejection at this stage at which the results are very premature for publication, with the possibility to resubmit in the future when the results are more coherent and substantial, and a more plausible mechanism is proposed.

First, we would like to thank the reviewer for providing several interesting comments and recommendation we used to improve our work. The activation system presented in this manuscript has never been described yet and the physical mechanism underlying the activation process is expected to be very complex since ionising radiations (IRs) generate many particle-matter interactions in water and understanding this process is still a cutting-edge research nowadays¹. For instance, there is still a lot of controversy in the literature about the mechanism of radioenhancement effect provided by metallic nanoparticles upon IRs^{2,3} even if this therapeutic approach has been described *in vivo* for the first time in 2004⁴ and is currently assessed in clinical trials for cancer treatment^{5,6}.

Nevertheless, we agree that the reviewer is expecting a better understanding about the activation process. Thus, we carried out several experiments for this purpose (cf. below). These new data revealed some key features: (i) validation of the key role of hydroxyl radicals using (i.a) new hydroxyl-radical quenching approaches, (i.b) conversion of hydrated electrons into hydroxyl radicals, and (i.c) chemical generation of hydroxyl radicals (Fenton chemistry); (ii) validation that the activation is oxygen-independent and (iii) evidence that activation proceeds without catalytic pathway.

Moreover, the pharmacological studies have been presented differently as suggested by the reviewer, and additional experiments were performed using flow cytometry. We agree that the cytotoxicity benefit is relatively low at this stage, but the main contribution of this manuscript is to present a new type of photoswitch activation by IR. The GdAzo compound used to prove this activation concept is certainly not the best candidate for medical applications (cf. discussion below), however we believe that such chemical-structure developments are out of the scope of this manuscript.

More specifically:

I would recommend a full language restructuring both from the gramatic/syntax point of view but also regarding the scientific terms. As an example "azote" is perfectly fine in French or Greek but is not used in English. Also the authors' favourite expression "lower quanta of energy" leaves many questionmarks regarding its meaning in most of the places used.

The term "azote" has been rectified and the two sentences containing "quanta of energy" have been modified: (i) "to locally convert the carried energy into lower quanta to induce" has been replaced by "to locally convert the carried energy into low-energy particles and species to induce" (l. 77-78, main text) and (ii) "release of lower quanta of energy in the very close vicinity" has been replaced by "the

release of low-energy particles and species in the very close vicinity" (l. 167, main text). Moreover, spelling, syntax and grammar have been checked to improve the writing quality of this manuscript.

Main points:

1. The title needs to be significantly toned down, as the authors suggest that they are the ones to break the photoactivation limit and with a clinical application. A brief review of the recent bibliography will show them that this limit has been broken before, whereas they must explain how exactly they adapted their work to clinical application. This applies to the whole of the text which should be restructured and toned down to match the findings; for example the introduction does not refer at all to prior attempts to activate photoactive compounds directly or indirectly by ionising radiation!!

Investigations to improve radiotherapy efficacy using co-injection of chemical entities first led to the development of nanoparticles able to increase the effect of the IR dose by radiosensitizing and/or radioenhancement effects, which was more recently associated to immune system activation due to the intrinsic immunogenicity of IRs^{7,8}. Even if this approach was rationalized by the increase in the amount of reactive oxygen species (ROSs) in the vicinity of the irradiated nanoparticles, there are current active discussions about understanding the underpinning mechanisms and pointing out all the contributions which are expected to rely on physical, chemical and biological effects^{3,9}.

In the last few years, several systems were reported to induce more complex and selective actions from IR stimuli^{10,11}. We could separate them into two main families depending on the type of activation based on down-conversion or oxidation by ROSs. The down-converting systems, such as

nanoscintillators, are designed to convert incident X-ray photons into UV-vis light to induce the release of cytotoxic agents such as singlet oxygen ($^1\text{O}_2$) in the case of the widely studied photodynamic therapy approaches¹². The other systems benefit from the generated ROSs (and/or potentially from the secondary electrons)¹³ to induce specific bond cleavage (mainly diselenide, disulfide, C-N, C-O, S-N, coordination bonds with metal)¹⁴⁻¹⁷, DNA break¹⁸, atom oxidation (mainly sulphur, selenium and carbon from unsaturated lipids)^{19,20} leading to the disassembly of capsules, polymers or prodrugs, and the release of drug or gas (nitric oxide, carbon monoxide).

The system GdAzo described in this manuscript differs from all these already-known X-ray-responsive technologies as it is based on the direct activation of a photoswitch, that is a molecular rearrangement (isomerisation) leading to a new configuration without neither inducing any bond cleavage nor disassembly. The current approach to activate photoswitch efficiently is to use lights that cannot penetrate deeply into tissues (UV to near-infrared wavelengths) which limits clinical application. Our approach can overcome this major limitation and we think that would be very valuable for next developments. Indeed, this could help to move the therapeutic systems based on photoswitch activation into wide clinical applications thanks to the high activation efficacy observed at low clinical doses (2-5 Gy) using different type of irradiation sources (photons or electrons at energy from 80 keV to 4.5 MeV). That is why we are mentioning in the title that we are using "ionising radiation stimuli adapted to clinical application". The title has been modified as follows: "Breaking photoswitch activation depth limit using ionising radiation stimuli adapted to clinical application" to be more specific on the activation system we describe here (photoswitch) and the text has been toned down to better fit with our results, as suggested by the reviewer. Moreover, a paragraph has been added to introduce the already-known technologies based on X-ray activation to induce specific pharmacological actions (l. 80-95).

2. In Fig. 2 we see a significant difference in the non-irradiated Azo or Gd Azo between 2 and 20 Gy . How do the authors explain that? This is a recurring theme also in Fig. 3 a and b... The non-irradiated Azo or GdAzo should be the same for 2 or 20 Gy, simply because this sample is not irradiated (0 Gy)!! What accounts for these sizable differences which are not covered by the error bars.

This difference is due to the thermal relaxation of the *cis*-GdAzo during the experiment. Indeed, the reviewer can find in the Supplementary Information section 5.1 (Quantification of *cis*-GdAzo activation upon ionising radiations, p. 25) the detailed protocol for this type of experiments. It is mentioned that the “the second microplate (plate 2) was irradiated upon incremental doses of ionising radiations (2, 3, 5 and 10 Gy)” and that “the full experiment was performed in about 4.5 h”. This is also mentioned in the main text (l. 188-189: “a conversion mainly attributed to thermal back relaxation of the *cis*-isomer in the dark”). This thermal relaxation was quantified using the control non-irradiated experiment and was corrected to the irradiated sample to obtain the activation efficacies (*G*-values) upon IR. Owing to the reviewer’s comment we have added a sentence to clarify this point in the Supplementary Information section 5.1 (p. 25, l. 422-423): “It has to be noted that the conversion observed for the non-irradiated compounds was due to thermal back relaxation.”.

3. In line 213 the authors based on their results they claim that the activation of *cis*-GdAzo is independent of the type of radiation and is the same for photons and electrons. How can they account for this? They try to produce a mechanism, but this does not agree with the results and is not fully substantiated. They attribute their system's action to HO· and associated oxidation, but: did they show that the same amounts of HO· are produced in all the cases of ionizing radiation? Why don't they use the well known HO· quencher mannitol to see the effect? The authors refer to the HO· conversion to H₂O₂, but how about the action of H₂O₂ on the *cis*GdAzo, have they checked it? And what about the reverse? How would the conversion of H₂O₂ to HO· affect the *cis*GdAzo? In other

words, would the Fenton reaction which takes place within the cells, lead to spontaneous activation of cisGdAzo?

And what happens to the cisGdazo activation in conditions of hypoxia? The fact that XR, GR and e- have the same activation effects remains a mystery.

The authors have a long road ahead to understanding and thus harnessing their system.

Indeed, the activation efficiency of *cis*-GdAzo is about the same with the 3 different sources used in our study: X-ray irradiator (photons, mean energy of 80 keV, in the 30-140 keV range, dose rate about 1 Gy/min), gamma-rays from Cesium-137 source (photons at 662 keV, dose rate about 1 Gy/min) and linear accelerator (Kinetron, electron at 4.5 MeV, dose rate about 4 Gy/min) and this is detailed in Supplementary Information section 5.4 p.62 (Determination of *G*-values and comparison of the radiation sources). Following the reviewer advice, we have carried out substantial new experiments in the suggested way to get new insights in the proposed mechanism. Experiments performed and results are as follows:

3.a Amount of hydroxyl radicals produced by the different sources:

First, we note that the 3 sources of radiation deliver low linear energy transfer (LET, which is defined by the rate of energy loss per unit length of track of the particle) radiations. The transient species produced by IRs in water have been well characterized in literature^{21,22}. In low-LET radiations, the first events appear in small widely separated spurs (10^{-16} - 10^{-10} s time scale) and the generated radicals are homogeneously spread into water at about 10^{-7} s after irradiation to yield aqueous electrons (0.28 $\mu\text{mol/J}$), hydrogen radicals (0.062 $\mu\text{mol/J}$), hydroxyl radicals (HO^\bullet , 0.28 $\mu\text{mol/J}$), hydrogen (0.047 $\mu\text{mol/J}$), hydrogen peroxide (H_2O_2 , 0.073 $\mu\text{mol/J}$) and protons (0.28 $\mu\text{mol/J}$). Nevertheless, the radiochemical yield of HO^\bullet can be affected by the energy of the incident particle²³, and can be lowered when low-energy photons (1-100 keV) are used. To answer this question, we

have quantified the amount of HO• generated by the X-ray generator (compared to the gamma-ray source) which delivers photons at energy affecting the HO• yield (cf. Fig. A, Fig. 5d (main text) and Supplementary Information section 9.1 p. 81, "Quantification of hydroxyl radicals and role of Gd", Figs. 59-63). The HO• production from the Kinetron and from the gamma-ray sources should be similar at the dose rate used (0.28 μmol/J)^{21,24}.

Indirect quantification of hydroxyl radicals (HO•) was based on HO• scavenging by coumarin²⁵ and quantification of 7-hydroxycoumarin (7-OH-Coum) which is the only fluorescent product released from this scavenging reaction²⁶. 7-OH-Coum production proved to be proportional to HO• concentration and specific for HO• among other reactive oxygen species². This quantification method seems to be robust and adaptable^{27,28}. Nevertheless, it has to be noted that this is an indirect method of quantification which can only give access to the amount of HO• available to react with coumarin in our experimental conditions (temperature, concentrations, etc.). In the conditions used, coumarin reacts with HO• ~100 ns after the initial transfer of energy to water (rate constant of this scavenging reaction is $k = 1.05 \cdot 10^{10} \text{ L} \cdot \text{mol}^{-1} \cdot \text{s}^{-1}$)²⁹ which results in quantification of the homogeneously distributed HO• without interfering on the intratrack recombination of radicals occurring in the spurs.

The radiochemical yields (*G*-values) of HO• determined in our study were 0.200 and 0.280 μmol/J upon X-ray and gamma-ray irradiations respectively (Fig. A, with [Gd³⁺]=0), which is in line with the reported yields for similar radiation sources and dose rates^{22,23}. Moreover, this study revealed that the presence of gadolinium in the solution increased the yield of HO• (enhancement factors of 33% at 200 μM [Gd³⁺] and 20% at 500 μM [Gd³⁺] upon X-ray and gamma-ray irradiations respectively).

Fig. A. Determination of $G(\text{HO}^\bullet)$ at different concentrations of Gd^{3+} . HO^\bullet production upon X-ray (XR) and gamma-ray (GR) in the presence of different concentrations of Gd^{3+} by considering a conversion yield of coumarin into 7-OH-Coum of 3.1% (from ref²). The experiments were repeated 5 times independently. The means \pm standard deviations are reported.

In this study, we showed that the amount of HO^\bullet was slightly lower from X-ray irradiation comparatively to the gamma-ray source, as expected from literature, even if the sources we used were all considered as low-LET radiations with a similar radiolytic energy deposition. Interestingly, the activation yield of *cis*-GdAzo (G -value) was also slightly lower upon XR compared to GR and E at 2-5 Gy (cf. Fig. 3d in the main text and Fig. 47 p. 63 in the Supplementary Information section 5.4), even if this difference was lower than what could be expected from the $G(\text{HO}^\bullet)$ difference.

3.b New HO^\bullet quenching experiments

tert-Butanol (tBuOH) has been used to convert HO^\bullet into much less reactive tertiary radicals resulting in quenching the oxidant properties of $\text{HO}^{\bullet 21}$. Nevertheless, we assessed the impact of two new HO^\bullet quenchers at different concentrations to validate our conclusion: mannitol and ethanol (EtOH) (Fig. B, and Supplementary Information section 7, Fig. 53 p. 73).

Fig. B. Inhibition of *cis*-GdAzo activation using HO[•] quenchers. Absorbance difference (365 nm) of medium containing *cis*-GdAzo compound (50 μM) before and after gamma-ray irradiation at different doses (triplicate). The compound was dissolved in MilliQ water (control) or in aqueous solution of *tert*-butanol (tBuOH, **a**, 10, 100, 524 mM (5% v/v)), mannitol (**b**, 10, 100, 524 mM) or ethanol (EtOH, **c**, 10, 100, 524 mM). The experiments were repeated 3 times independently. The means ± standard deviations are reported. OD: optical density.

A similar impact of tBuOH, mannitol and EtOH on *cis*-GdAzo activation was observed. The activation was clearly inhibited by these species known to quench the oxidant property of HO^{•21}. This quenching effect was dependant on the quencher concentration for the higher radiation doses (10-20 Gy).

3.c Investigations on H₂O₂ and Fenton-type activation

The chemical induction of *cis*-GdAzo activation has also been investigated and is detailed in the Supplementary Information section 8.2 p. 79 (*cis*-GdAzo activation using Fenton chemistry). We first carried out absorbance measurement to assess a potential activation^{30,31}. While hydrogen peroxide (H₂O₂) alone was not able to activate *cis*-GdAzo, Fenton chemistry induced this activation (Fig. Ca). The Fenton activation was confirmed by HPLC with an increase in *trans*-GdAzo from 16.7% to 48.4%

(Fig. Cb), even if some degradation of GdAzo was also detected (14.8%) (Figs. 5a, 5b in the main text and Fig. 58 p. 80 in Supplementary Information section 8.2).

Fig. C. Chemical activation of *cis*-GdAzo by H₂O₂ and Fenton chemistry. **a**, Absorbance of *cis*-GdAzo (photostationary state at time 0) recorded 5 min after addition of H₂O₂ only (50 mM, blue squares) or after running Fenton reaction (red triangles) (Control in water) (n=3). **b**, Proportion of *trans*-GdAzo after running Fenton reaction (Control without Fenton reagents) in *cis*-GdAzo-containing medium (photostationary state at time 0, just before addition of H₂O₂) determined by HPLC (n=3). The means ± standard deviations are reported. OD: optical density.

3.d Investigations on activation in controlled atmosphere

To study *cis*-GdAzo activation in conditions of hypoxia, we carried out irradiation under controlled atmosphere. These experiments are detailed in the Supplementary Information section 8.1 p. 77 (*cis*-GdAzo activation under N₂ and N₂O gas saturation).

First, we demonstrated by HPLC that absence of oxygen (N₂ saturation) was not affecting at all *cis*-GdAzo activation (Fig. Da and Fig. 4c in the main text). This oxygen-independent type of activation could be valuable for hypoxic-tumor treatment. Moreover, this bears out that hydrated electrons and hydrogen radicals (and thus the couple perhydroxyl radical/superoxide radical anion HO₂[•]/O₂^{•-} resulting from their reaction with oxygen) are not involved in *cis*-GdAzo activation otherwise absence of oxygen would have lowered activation efficacy.

Secondly, *cis*-GdAzo activation was quantified in solution saturated with nitrous oxide (N₂O) which converts all hydrated electrons into HO• upon irradiation of aqueous solution, resulting in doubling the production yield of HO• (thus $G(\text{HO}^\bullet) = 0.56 \mu\text{mol}/\text{J}$, completed in $\sim 14 \text{ ns}$). As expected, N₂O saturation led to an increase in *cis*-GdAzo activation efficacy (Fig. Da and Fig. 4c in the main text). Furthermore, when the irradiation dose in N₂O saturated solution was brought back to the generated HO• amount, the molecular activation of *cis*-GdAzo was similar as in water (Fig Db and Fig. 4d in the main text). For instance, 5 Gy irradiation in N₂O-saturated solutions (equivalent to 10 Gy in non-saturated solutions with respect to generated HO• amount, and noted “eq 10” in Fig. Db) resulted in similar activation efficacy as 10 Gy irradiation in non-saturated solutions.

Fig. D. Molecular activation of *cis*-GdAzo upon gamma-ray radiation under N₂ and N₂O gas

saturation. a, Molecular activation of *cis*-GdAzo (50 μM , H₂O) upon gamma rays in N₂ or N₂O-saturated solutions (Control without inerting) determined by HPLC and reported as the difference of *trans*-isomer proportion before and after gamma-ray irradiation ($n=3$). **b,** Comparison of *cis*-GdAzo activations in control (no inerting) and N₂O-saturated solutions in conjunction with the equivalent (eq.) amount of hydroxyl radicals generated depending on doses. N₂O saturation generates two times more hydroxyl radicals at the same dose, thus “eq 4”, “eq 10”, “eq 20” and “eq 40” relate to the doses of 2, 5, 10 and 20 Gy in N₂O-saturated solutions. The means \pm standard deviations are reported.

3.e Investigations at different concentrations of GdAzo

In the first version of the manuscript, we suggested a hypothesis for explaining the *cis*-GdAzo activation mechanism, based on a catalytic effect (the oxidised *trans*-GdAzo^{**} was reduced by *cis*-GdAzo). We tried to validate this hypothesis by investigating *cis*-GdAzo activation upon gamma-ray irradiation at different *cis*-GdAzo concentrations (20, 200, 500 and 1000 μ M, to be added to the 50 μ M concentration, as already assessed in the first version of the manuscript) (Fig. E). This additional piece of experiments has been fully described in the Supplementary Information section 10 p. 87 (Investigation of activation mechanism by variation of GdAzo concentration).

Fig. E. Molecular activation of *cis*-Azo and *cis*-GdAzo at different concentrations upon gamma-ray radiation. Molecular activation of *cis*-GdAzo and *cis*-Azo (control molecule without Gd atom) at concentration of 20 μ M (a), 200 μ M (b), 500 μ M (c) and 1000 μ M (d) determined by HPLC and reported as the difference of *trans*-isomer proportion before and after gamma-ray (GR) irradiation (n=3). The means \pm standard deviations are reported. Two-way Anova (Bonferroni post-test) was used for statistical analyses (All vs GdAzo GR). *** $P < 0.001$.

Thanks to those additional experiments, we could now determine the relation between the radiochemical yield (G -value) of *trans*-GdAzo and the initial concentration of *cis*-GdAzo. The linear relation observed (Fig Fa) shows that there is no catalytic effect for *cis*-GdAzo activation, which has the same efficacy for low and high *cis*-GdAzo concentration. Moreover, we observed that $G(\textit{trans}\text{-GdAzo})$ (*trans*-GdAzo production per unit energy) was higher for lower doses which reveals a larger loss of energy at higher doses (excess of HO^\bullet induces other non-specific reactions or recombination processes). This study allowed us to establish a predictive model to estimate $G(\textit{trans}\text{-GdAzo})$ from the initial *cis*-GdAzo concentration and the radiation dose since $G(\textit{trans}\text{-GdAzo})$ follows a logarithmic decrease when the dose increases in the range 2-20 Gy (Fig Fd).

Fig. F. Relation of G -values vs *cis*-GdAzo and *cis*-Azo concentrations upon gamma-ray radiation.

a, b, G -value ($\mu\text{mol/J}$) of the activation of *cis*-GdAzo (**a**) and *cis*-Azo (**b**) upon gamma ray determined by HPLC (molar amount of *trans*-GdAzo and *trans*-Azo activation corrected from thermal back relaxation) (3 independent experiments). **c**, Parameters obtained from linear regressions in **a** and **b**, using the least-squares method (Excel 2016). **d**, Linear regression to correlate $G(\textit{trans}\text{-GdAzo})/[\textit{cis}\text{-GdAzo}]$ to the irradiation doses. The relation $Y = -0.036 \cdot \ln(X) + 0.1304$ (with Y the slope of linear

regressions from a in the unit $G\text{-value}/\mu\text{M GdAzo}$ and X the dose in Gy) was obtained with $r^2 = 0.9925$ (Excel 2016).

3.e Conclusion

This bunch of experiments sheds a new light on the underlying mechanism of *cis*-GdAzo activation upon ionising radiations. First, the preponderant role of the hydroxyl radicals HO^\bullet has now been validated using several approaches including new quenchers (tBuOH, mannitol and EtOH at different concentrations), N_2O -saturated solutions (conversion of hydrated electrons into HO^\bullet) and chemical activation using Fenton chemistry. We also have directly demonstrated that others species such as hydrated electrons, hydrogen radicals and hydrogen peroxide do not interfere in the activation process thanks to experiments performed in N_2 -saturated solutions or in the presence of hydrogen peroxide. These new experiments, in addition to the previous ones (reported in the last version of the manuscript), clearly validate that HO^\bullet is the key specie that contributes to *cis*-GdAzo activation.

Furthermore, accurate investigation on the quantification of HO^\bullet from the X-ray and gamma-ray sources showed a difference in the production yield $G(\text{HO}^\bullet)$ between these two sources (0.200 vs 0.280 $\mu\text{mol}/\text{J}$ respectively). The activation yield of *cis*-GdAzo was slightly lower upon XR compared to GR and E at 2-5 Gy, even if this difference was lower than what could be expected from the $G(\text{HO}^\bullet)$ difference.

Finally, the catalysis mechanism hypothesised in first instance has been investigated and was then declined by the new bunch of experiments carried out at different concentrations of *cis*-GdAzo. Moreover, these experiments led to implement a relation between the activation yield $G(\textit{trans}\text{-GdAzo})$ and the radiation dose in the 2-20 Gy range and showed a better efficiency at lower dose.

The similar activation efficacy observed with the 3 different sources used in this study points out that the interaction of the first incident particles is not important in this process and that secondary species (HO^{*} here) play a key role. However, water radiolysis generates many events, including both reductive and oxidative processes and the understanding of these events are still intensively studied and remain prone to discussion (for instance, cf. ref¹ or ref³²). This study provides several elements and demonstrations that result in a better understanding of the implemented processes during *cis*-GdAzo activation and the authors acknowledge the excellent reviewer suggestions for mechanism clarification. However, we are conscious that the accurate mechanism underlying these complex processes will still require many efforts to be fully unlocked. Indeed, illustration of the complexity of ionising-radiation chemistry is revealed by the current discussions to fully understand the mechanisms of radioenhancement effect induced by metallic nanoparticles^{3,9}, even though this research field was born two decades ago⁴.

4. The authors need to identify how deeply into the membrane their molecule digs, since the hydrophilic head pulls the molecule out of the membrane. I do not agree with the authors extrapolations from Fig. 4. Firstly, one glance at the big error bars is enough to see that in Fig. 4g there is no statistical significance (difference) between the white(NI) and black (GR) bars. From 4h we can see that the compound confers a significant toxicity on its own (70%) at 850uM and if we consider the negative error bar of 0 concentration for the black bar (irradiated)then we see that irradiation could cause a ~70% toxicity, which makes the 250 uM and 500 uM results non significant (the 500uM one was non-significant anyway because of the huge white bar error). That leaves the case of the 850uM where the compound chemical toxicity is already 70%. Similar extrapolations can also be made for the case of gem. Also about gem the authors state that the amount of gem does not increase in gem-resistant cells in the presence of trans-GdAzo. But if trans-GdAzo indeed

permeabilizes the membranes (as the authors claim) to the point that PI can flood in, then why does not gem go in through the permeabilized membranes?

Molecule – membrane interaction:

SAXS and DSC experiments demonstrate that *trans*-GdAzo interacts with DPPC bilayers, leading to a modification of their structural and thermal parameters (Supplementary Information section 13.3, p. 122). The location of *trans*-GdAzo in phospholipid bilayers was further addressed by comparing the electron density profiles of bilayers of DPPC, on one hand, and DPPC with 0.1 mol% *trans*-GdAzo on the other hand (Fig. G and Fig. 6b in the main text). The DPPC profile reflected the electron-rich phosphatidylcholine headgroups and electron-poor hydrocarbon chains, in agreement with previous studies³³. The distance between the phosphate groups, d_{HH} , deduced from the position of the two maxima was 44.0 Å in pure DPPC. The addition of *trans*-GdAzo slightly shifted these maxima away from the bilayer centre ($d_{HH}' = 46.4$ Å) and increased their intensity. These results are consistent with the insertion of *trans*-GdAzo between DPPC chains, with the electron-rich Gd-chelate moiety slightly protruding in water. Of note, the thickness of the bilayer hydrophobic core ($2D_C = 34.4$ and 28.5 Å at 20 and 50 °C respectively)³³ enables the insertion of the hydrophobic part of *trans*-GdAzo whose length can be estimated at 21.8-22.6 Å, according to the molecular model (cf. Supplementary Information section 12).

Fig. G. Insertion of *trans*-GdAzo in DPPC model membrane. Electronic density of the DPPC model phospholipid membrane in the presence of *trans*-GdAzo (0.1%) at room temperature extracted from SAXS experiments (black: $x_{trans-GdAzo} = 0\%$ (pure DPPC), red: $x_{trans-GdAzo} = 0.1\%$).

Cell permeabilisation:

The wide standard deviation observed from the microscopy experiments (Fig. 4g in the first version of manuscript text) is due to the large variation of the extreme data points. These data were represented differently using median with interquartile range, which is a robust representation of data dispersion that lowers the visual impact of the extreme data values (Fig. 7e of the main text). We concluded on the significant difference between NI vs GR for some conditions (mentioned on the figure) using two-way Anova (Bonferroni post-test, matched by rows). Moreover, to confirm that *cis*-GdAzo induces cell permeabilisation upon IRs, we carried out an additional experiment using flow cytometry (cf. Fig. H, Fig. 7f-7h (main text) and Supplementary Information section 14.4 p. 141) on the human T lymphocyte cell line used for the cytotoxicity experiment (CCRF-CEM ARAC-8C). These new experiments clearly confirmed the pharmacological effect (i.e., cell permeabilisation) previously observed using microscopy.

Fig. H. Cell membrane permeabilisation induced by *cis*-GdAzo upon gamma-ray irradiation using flow cytometry. Quantification of cell permeabilisation (CCRF-CEM ARAC-8C) after treatment with *cis*-GdAzo upon gamma-ray irradiation (GR, 2 Gy) at 15 min (a), 30 min (b) and 45 min (c) after irradiation (n=3). The relative increase in PI-positive events compared to medium without *cis*-GdAzo is represented. The means \pm standard deviations are reported. Two-way Anova (Bonferroni post-test) was used for statistical analyses. ns: not significant * $P < 0.1$, ** $P < 0.01$. *** $P < 0.001$. NI: non irradiated.

Cytotoxic effect:

Indeed, we have reported the difference between irradiated and non-irradiated cells in the presence of *cis*-GdAzo. As suggested by the reviewer, we analysed the influence of the *cis*-GdAzo concentration on the cell viability for the different cell treatments (GR and GR + Gem). These analyses have been added in the Supplementary Information (section 14.6, Figs. 90 p. 145) and are presented below (Fig. I). As mentioned above, the statistical tests were performed using two-way Anova (Bonferroni post-test, matched by rows) and were mentioned on the figure caption. We could conclude that the addition of *cis*-GdAzo (for the three concentrations tested) significantly reduced cell viability upon GR (*vs* absence of *cis*-GdAzo). Nevertheless, this was not the case in the presence of Gem except for the highest *cis*-GdAzo concentration (850 μ M), which is in agreement with the absence of additional cytotoxicity arising from Gem, as already mentioned in the main text (l. 558).

Fig. 1. Assessment of the influence of *cis*-GdAzo concentration on cell viability. Cell viability (CCRF-CEM ARAC-8C) in the absence or presence of *cis*-GdAzo at concentrations of 250 μ M (a), 500 μ M (b) and 850 μ M (c) in different conditions: no irradiation (NI), NI in the presence of Gem (0.1 μ M), 2 Gy gamma-ray (GR) irradiation and 2 Gy GR irradiation in the presence of Gem (0.1 μ M) (n=3). The data are represented as the means \pm standard deviations. Two-way Anova (Bonferroni post-test) was used for statistical analyses (absence vs presence of *cis*-GdAzo). ns: not significant ($P > 0.05$), *: $P < 0.05$, **: $P < 0.01$, ***: $P < 0.001$.

Gem internalization:

The radiolabelling experiment (using [3 H]Gem) showed that Gem was not internalized into cells in the presence of *trans*-GdAzo. This observation is in agreement with our conclusions drawn from the cytotoxicity experiment which revealed that Gem was not significantly impacting the cell viability in the presence of *cis*-GdAzo upon IR (Supplementary Information section 14.6, Fig. 91 p. 146). We discussed in the main text of the manuscript that *cis*-GdAzo upon IRs induced a loss of integrity for some cells leading to collapse (Fig. 7I in the main text and Supplementary Videos), which was attributed to the partial solubilisation of the phospholipid bilayers as shown by SAXS on model membrane (Fig. 6c in the main text and Supplementary Information section 13). The propidium iodide (PI) penetrates inside dead cells, goes to the nucleus and binds to double stranded DNA. This label of dead cells is quite persistent because cell integrity is impacted. The dead cells could not accumulate Gem, which can be explained by the inability of these cells to phosphorylate Gem or by

the collapse of the cells before and during the centrifugation steps which were required to wash the cells before scintillation counting. This explanation has been added into the main text (l. 561-562).

5. PI is not very reliable under prolonged incubation as it might stain the cells even without cell permeabilization. Figs. 4BC and 4DE are shown in different magnification modes. It is difficult to compare the extent of PI signal between the groups. Those panels of the same magnification are necessary to provide. The same situation in Supplementary Information sections 11.2 and 11.3.

The experiments using PI as a biomarkers were carried out for maximum 1 h period, which is quite standard for this type of cell labelling³⁴. It is true that PI staining may be considered as questionable for monitoring the viability of bacteria, but it is a completely different application than the approach followed in our study³⁵⁻³⁷. Noteworthy, unspecific PI labelling was monitored during control experiments (with PI and without neither *cis*-GdAzo nor IR) and was negligible in our experimental conditions (for instance cf. Supplementary Information section 14.2, Fig. 79, p. 131). Besides, experiments based on cell counting (to determine the cell viability) have been carried out to confirm the cytotoxicity of the *cis*-GdAzo upon IR (Figs. 7j, 7k in the main text). Finally, the magnification of Figs. 7a-7b (main text) has been modified to get a magnification similar as in Figs. 7c, 7d (main text). These new images have also been introduced in the Supplementary Information section 14.2 (Figs. 79-85, p. 131-137).

6. it seems that the GdAzo-caused enhancement in cell killing was quite low without even taking the error bars into account. If one applies simple mathematics to the data presented in Fig. 4h and crude numbers taken from the presented figure in then only 10% enhancement can be obtained:

$$(\text{NIGdAzo} - \text{GRGdAzo}) - (\text{Ctrl} - \text{GR}) = 80 - 25 - (100 - 55) = 55 - 45 = 10\%$$

This is too low a synergy also presented in the showcase results. Perhaps it would be a good idea for the authors to present their results in different experiments so the median gain from the use of *cis*GdAzo can become clearer.

We agree with the reviewer that we cannot conclude to a synergistic effect according to the data presented. We focused here on the demonstration of a cytotoxic effect compared to control experimental conditions (no irradiation or no *cis*-GdAzo). The *cis*-GdAzo compound used in our study enabled to describe a new concept, but it still need to be optimized (modification of substituents on the aryl cycles for instance) in order to improve and adjust the therapeutic index for further preclinical development.

7. How does the molecule (Gd-Azo) exist more stably in its natural form? Is it in the *cis* form, or in the *trans* form? The understanding is that the *cis* form does not kill but the *trans* one kills. If the molecule originally exists in the the *cis* form then disregard the question. However if the compound naturally exists more stably in the *trans* form (as stated in the manuscript text) and the authors needed to switch it to *cis* by UV light, before giving it to cells and applying the radiation to switch it again to *trans* then this raises many questions! That is because in any case the *cis* compound will thermodynamically convert to *trans* within 12-24h (according to the authors) and will kill the cells anyway! perhaps this is not a disaster in the case of cell cultures (in vitro) where one can remove the compound before the thermodynamic restructuring , but it would be a disaster in the case of the patient, where if the compound is not fully metabolised and excreted in 12-24h then there will be a big spontaneous toxicity problem. The authors need to clarify their use of the compound throughout the manuscript text, and make it easier to comprehend for the reader as it currently is a bit obscure.

The *trans*-GdAzo is the stable isomer, and the *cis*-GdAzo is metastable. As mentioned by the reviewer the metastable and inactive *cis*-GdAzo is converted into the active *trans*-isomer over time in the dark by thermal relaxation. The compound presented in this paper has a relatively short thermal back relaxation (Supplementary Information section 3.2, Fig. 14, p. 23, $t_{1/2} = 2.3$ h at 37 °C in PBS) and has been used to prove the activation approach upon IR. Indeed, this compound can be used for *in vitro* experiment (in a limited time scale), however, there is no doubt that further optimisations of the chemical structure is required before any translation to *in vivo* study. Nevertheless, several approaches have already been described in literature to slower thermal back relaxation of azobenzene derivatives up to several months/years³⁸. We estimate that a thermal half-life of few weeks should be adapted according to previous biodistribution studies with Gd-chelate-based ultrasmall nanoparticles (< 5 nm) which exhibited a plasma half-life of about 1.5 h in human.⁶ A sentence has been added to clarify this point for the reader: "It has to be noted that the thermal half-life of *cis*-GdAzo is well adapted for *in vitro* experiments, however it should be increased to several days or weeks for *in vivo* investigation" (l. 154-156 in the main text).

8. Lines 113-115. The authors supposedly refer to other Azo derivatives absorbing in near-infrared region as Fig. 2B shows absorption up to about 500nm. That must be clarified because excitation wavelength in the setup of confocal microscopy falls into that region.

We referred to azobenzene derivatives absorbing in near-infrared region to introduce the azobenzene moiety in the text ("Azobenzene is known for reversible shift between *trans* and *cis* configurations through isomerisation of the azo double bond after excitation by photons from UV to near-infrared energies" (l. 142-144 main text). GdAzo compound does not absorb in the near-infrared region (cf. Fig. 2b in the main text for instance) since negligible absorption was detected above 525 nm. During the microscopy experiments, propidium iodide was detected using excitation

wavelengths at 538 nm (cf. Supplementary Information section 14.2 p. 130) or 561 nm (cf. Supplementary Information section 14.3 p. 138) which were adapted as evidenced by the control experiments without GdAzo compound (cf. Supplementary Information section 14.2, Fig. 79, p. 131 for instance) .

minor comment

Line 74-75: The authors refer to all ionizing radiation as particles and while this is more correct for electrons or protons it is not entirely correct for photons.

We agree that all particles described by quantum mechanics can be designated as either a particle or a wave. However, we think that it is quite reasonable to use the adapted definition of light depending on the type of experiment carried out, and to consider light as a wave for propagation and as a particle for interaction with matter (which is what we focused on in this work). Nevertheless, we modified this sentence to avoid any confusion: "Our approach is to use high-energy waves/particles contained in ionising radiations..." (l. 74-75 in the main text).

Reviewer #2 (Remarks to the Author):

This is a great paper. It makes a very substantial new contribution to the application of photochemistry to biomedicine. It is very thorough and represents a substantial amount of work. The central idea is that ionizing radiation, which can penetrate deep into tissues, can lead to the production of an oxidant (e.g. an OH radical) by a Gd chelate. If this occurs close to a cis-azobenzene (e.g. covalently linked to the Gd chelate), it can trigger thermal cis to trans isomerization of the azobenzene through a N=N+ azo radical. This builds substantially on conceptual work by Hecht (as referenced), and convincingly demonstrates that cis-to-trans azo isomerization can be done with ionization radiation in practice. I am convinced by the extensive data on the compounds analyzed in solution. To my mind, this is enough of a conceptual, and practical advance to warrant publication in Nature. The authors take the work a bit farther, however, and analyze cell permeabilization by the azo-linked Gd chelate, likely via a membrane disruption mechanism. While this data does seem to show an effect, I would be more convinced if flow cytometry data were used to quantitatively measure degree of permeabilization, rather than imaging data. In any case, to my mind, this is likely not the compound that would actually be employed in a clinical setting. Instead chemists and photomedicine experts can begin to design ideal compounds now that the photoactivation depth limit has been broken! One thing that would make the work even more compelling is if the authors could inject the cis compound into a mouse and recover the trans after irradiation! But perhaps this is too difficult if the thermal half life is 2-3h anyway (at room temp?). Congratulations!

Andrew Woolley

The authors thank very much the reviewer for this enthusiastic comment on their study.

As proposed by the reviewer, additional cell-membrane permeabilisation experiments have been investigated using flow cytometry (cf. Fig. J, Fig. 7f-7h (main text) and Supplementary Information section 14.4 p. 141, Figs. 88-89) on the human T lymphocyte cell line used for the cytotoxicity experiment (CCRF-CEM ARAC-8C). These new experiments clearly confirmed the pharmacological effect previously observed using microscopy.

Fig. J. Cell membrane permeabilisation induced by *cis*-GdAzo upon gamma-ray irradiation using flow cytometry. Quantification of cell permeabilisation (CCRF-CEM ARAC-8C) after treatment with *cis*-GdAzo upon gamma-ray irradiation (GR, 2 Gy) at 15 min (a), 30 min (b) and 45 min (c) after irradiation (n=3). The relative increase in PI-positive events compared to medium without *cis*-GdAzo is represented. The means ± standard deviations are reported. Two-way Anova (Bonferroni post-test) was used for statistical analyses. ns: not significant * $P < 0.1$, ** $P < 0.01$. *** $P < 0.001$. NI: non irradiated.

As understood by the reviewer, the compound described in this work still needs chemical modifications to optimise the photophysical properties for further *in vivo* studies. Indeed, the thermal half-life of *cis*-GdAzo ($t_{1/2} = 2.3$ h at 37 °C) remains too low to perform any relevant *in vivo* investigation since the conversion into the toxic isomer would happen in animals in a quite short time scale, and thus this raises also some ethical questions. We decided to first develop more efficient compounds before starting further *in vivo* investigations.

Reviewer #3 (Remarks to the Author):

The paper 'Breaking photoactivation depth limit using ionising radiation stimuli adapted to clinical application' by Bort et al. describes the application of ionizing radiation to induce isomerization of azobenzene conjugated to Gd-DOTA complex. The paper is clearly very intriguing and as such it should be of interest to Nature Communications. However, I think that the paper should be accepted only when the mechanism behind this isomerisation is properly proven, which is not the case yet. I have the following comments/suggestions.

First, we would like to thank very much the reviewer for his/her time to help us for providing a better understanding of the mechanism behind the isomerisation. Thus, we have performed a new bunch of experiments following the reviewer recommendations, as well as, other additional experiments to push forward the understanding in the underlying mechanism of *cis*-GdAzo activation process.

- The authors explain that high Z materials have better adsorption of high energy radiation, such as gamma's and X-rays. This is certainly true but the papers the authors refer to provide experiments using nano-particles made of high Z materials while they use Gd-DOTA complexes, hence much less Gd ions are probably present. The interaction probability in this case will depend on the concentration of Gd, something that is entirely not discussed and also not mentioned in the simulation part.

Some of the nanoparticles currently used for radioenhancement are indeed made of metal (oxidation state of 0) such as gold³⁹ or platinum⁴⁰, or metal oxide such as hafnium oxide⁵. However, ultra-small

nanoparticles constituted of gadolinium chelates (oxidation state of +3, 10-15 Gd-DOTA complexes aggregated on a 3-5 nm-size particle) have also been described as efficient radioenhancer tools in several preclinical models⁴¹. This type of nanoparticle has already been validated in a clinical trial of phase Ib⁶ and is currently assessed in phase 2 (NANORAD2, NCT03818386) and other phase 1 trials, which confirms that nanoparticles made of assembled Gd-chelate can also provide such radioenhancement effect. A reference of this type of nanoparticle has been introduced in the main text⁴². GdAzo is able to aggregate into ellipsoidal micelles of about 2.5-3.7 nm (cf. Supplementary Information section 11) which should assemble several Gd-DOTA complexes similarly. Moreover, as mentioned by the reviewer, the local concentration of Gd is probably influencing interactions with ionising radiations and this comment has now been added in the main text of the manuscript (l. 104-106).

• I am also not entirely convinced by the use of scavengers. Why use NaN₃, this is a typical scavenger for Singlet Oxygen? Many of the used scavengers may influence also the production/elimination rate of other species. Butanol for instance also affects hydrogen peroxide presence. The scavenging experiments are actually much more complex than presented in the paper.

We agree with the reviewer that each scavenger has some limitations (due to the lack of specificity, release of by-products, induction of side reactions, etc.) and that we should not draw any definitive conclusion from experiments using only one type of scavenger. However, our approach was to use a pool of several types of scavengers interacting with different species (hydroxyl radicals (HO[•]) and hydrated electrons mainly) and generating different side effects. We have tried to discuss these aspects in details in the Supplementary Information section 7 p. 69 (Investigation of activation mechanism using various converter species). In the first version of the manuscript, we used 5 different scavengers (*tert*-butanol (tBuOH), sodium azide (NaN₃), dimethylsulfoxide, cadmium

perchlorate and sodium selenate) at different concentrations. NaN_3 is, indeed, usually used to scavenge singlet oxygen (to study photodynamic therapy for instance), but it is also very useful to convert HO^\bullet into azide radicals (N_3^\bullet) which undergo selective electron transfer oxidation contrary to HO^\bullet ^{21,43}. Contrary to N_3^\bullet , HO^\bullet induces other types of reactions than electron transfer such as hydrogen atom abstraction (by interacting with hydrogen radical (H^\bullet) or by removing H^\bullet from a substrate), addition to unsaturated bond, interaction with an hydrated electron or with another HO^\bullet radical to make HO^- or H_2O_2 respectively. It is also true that *t*BuOH will affect the presence of hydrogen peroxide due to the scavenging of HO^\bullet .

Our study showed that *cis*-GdAzo activation was quenched by *tert*-butanol which suggested that HO^\bullet was the species leading to the activation process. This conclusion was not denied from the results obtained with the 4 other scavengers. Nevertheless, the demonstration of the preponderant role of HO^\bullet is at the core of understanding the activation mechanism and so we carried out several additional experiments, as requested by the reviewer, to validate this conclusion using the following approaches: (a) using new scavengers, (b) chemical activation by HO^\bullet using Fenton chemistry and (c) using gas saturation.

3.a New HO^\bullet quenching experiments

*t*BuOH has been used to convert HO^\bullet into much less reactive tertiary radicals resulting in quenching the oxidant properties of HO^\bullet . Nevertheless, we assessed the impact of two new HO^\bullet quenchers at different concentrations to validate our conclusion: mannitol and ethanol (EtOH) (Fig. K and Supplementary Information section 7, Fig. 53 p. 73).

Fig. K. Inhibition of *cis*-GdAzo activation using HO[•] quenchers. Absorbance difference (365 nm) of medium containing *cis*-GdAzo (50 μM) before and after gamma-ray irradiation at different doses (triplicate). The compound was dissolved in MilliQ water (control) or in aqueous solution of *tert*-butanol (tBuOH, **a**, 10, 100, 524 mM (5% v/v)), mannitol (**b**, 10, 100, 524 mM) or ethanol (EtOH, **c**, 10, 100, 524 mM). The experiments were repeated 3 times independently. The means ± standard deviations are reported. OD: optical density.

A similar impact of tBuOH, mannitol and EtOH on *cis*-GdAzo activation was observed. The activation was clearly inhibited by these species known to quench the oxidant property of HO^{•21}. This quenching effect was dependant on the quencher concentration for the higher radiation doses (10-20 Gy).

3.b Investigations on H₂O₂ and Fenton-type activation

The chemical induction of *cis*-GdAzo activation has also been investigated and is detailed in the Supplementary Information section 8.2 p. 79 (*cis*-GdAzo activation using Fenton chemistry). We first carried out absorbance measurement to assess a potential activation^{30,31}. While hydrogen peroxide (H₂O₂) alone was not able to activate *cis*-GdAzo, Fenton chemistry induced this activation (Fig. La). The Fenton activation was confirmed by HPLC with an increase in *trans*-GdAzo from 16.7% to 48.4%

(Fig. 1b), even if some degradation of GdAzo was also detected (14.8%) (Figs. 5a, 5b in the main text and Fig. 58 p. 80 in Supplementary Information section 8.2).

Fig. 1. Chemical activation of *cis*-GdAzo by H₂O₂ and Fenton chemistry. **a**, Absorbance of *cis*-GdAzo (photostationary state at time 0) recorded 5 min after addition of H₂O₂ only (50 mM, blue squares) or after running Fenton reaction (red triangles) (Control in water) (n=3). **b**, Proportion of *trans*-GdAzo after running Fenton reaction (Control without Fenton reagents) in *cis*-GdAzo-containing medium (photostationary state at time 0, just before addition of H₂O₂) determined by HPLC (n=3). The means \pm standard deviations are reported. OD: optical density.

3.c Investigations on activation in controlled atmosphere

To validate the preponderant role of HO[•] for *cis*-GdAzo activation, we carried out irradiation under controlled atmosphere. These experiments are detailed in the Supplementary Information section 8.1 p. 77 (*cis*-GdAzo activation under N₂ and N₂O gas saturation).

First, we demonstrated by HPLC that the absence of oxygen (N₂ saturation) was not affecting at all *cis*-GdAzo activation (Fig. 1a, and Fig. 4c in the main text). This bears out that hydrated electrons and hydrogen radicals (and thus the couple perhydroxyl radical/superoxide radical anion HO₂[•]/O₂^{•-} resulting from their reaction with oxygen) are not involved in *cis*-GdAzo activation, otherwise the absence of oxygen would have lowered activation efficacy. Moreover, this oxygen-independent type of activation could be valuable for hypoxic-tumor treatment.

Secondly, *cis*-GdAzo activation was quantified in solution saturated with nitrous oxide (N₂O), which converts all hydrated electrons into HO• upon irradiation of aqueous solution, resulting in doubling the production yield of HO• (thus $G(\text{HO}^\bullet) = 0.56 \mu\text{mol/J}$, completed in $\sim 14 \text{ ns}$). As expected, N₂O saturation led to an increase in *cis*-GdAzo activation efficacy (Fig. Ma, and Fig. 4c in the main text). Interestingly, when the irradiation dose in N₂O-saturated solution was brought back to the generated HO• amount, the molecular activation of *cis*-GdAzo was similar as in water (Fig Mb and Fig. 4d in the main text). For instance, 5 Gy irradiation in N₂O-saturated solutions (equivalent to 10 Gy in non-saturated solutions with respect to generated HO• amount, and noted “eq 10” in Fig. Mb) resulted in similar activation efficacy as 10 Gy irradiation in non-saturated solutions.

Fig. M. Molecular activation of *cis*-GdAzo upon gamma-ray radiation under N₂ and N₂O gas

saturation. a, Molecular activation of *cis*-GdAzo (50 μM , H₂O) upon gamma rays in N₂ or N₂O-saturated solutions (Control without inerting) determined by HPLC and reported as the difference of *trans*-isomer proportion before and after gamma-ray irradiation (n=3). **b**, Comparison of *cis*-GdAzo activations in control (no inerting) and N₂O-saturated solutions in conjunction with the equivalent (eq.) amount of hydroxyl radicals generated depending on doses. N₂O saturation generates two times more hydroxyl radicals at the same dose, thus “eq 4”, “eq 10”, “eq 20” and “eq 40” relate to the doses of 2, 5, 10 and 20 Gy in N₂O-saturated solutions. The means \pm standard deviations are reported.

3.d Conclusion

This bunch of experiments sheds a new light on the underlying mechanism of *cis*-GdAzo activation upon ionising radiations. The preponderant role of the hydroxyl radicals HO• has now been validated using several approaches including new quenchers (tBuOH, mannitol and EtOH at different concentrations), N₂O-saturated solutions (conversion of hydrated electrons into HO•) and chemical activation using Fenton chemistry. We also have directly demonstrated that other species, such as hydrated electrons, hydrogen radicals and hydrogen peroxide do not interfere in the activation process, thanks to experiments performed in N₂-saturated solutions or in the presence of hydrogen peroxide. These new experiments, in addition to the previous ones (reported in the last version of the manuscript), clearly validate that HO• is the key specie that contributes to *cis*-GdAzo activation.

• The most intriguing part is the lack of influence of radiation type. The same effects are observed for gamma's, X-rays and electrons. I miss here the G values for OH radical for each radiation source. The experiments suggest that the same amount of OH radicals are produced. Is that true?

Indeed, the activation efficiency of *cis*-GdAzo is about the same with the 3 different sources used in our study: X-ray irradiator (photons, mean energy of 80 keV, in the 30-140 keV range, dose rate about 1 Gy/min), gamma-rays from Cesium-137 source (photons at 662 keV, dose rate about 1 Gy/min) and linear accelerator (Kinatron, electron at 4.5 MeV, dose rate about 4 Gy/min). This study is detailed in Supplementary Information section 5.4 p. 62 (Determination of G-values and comparison of the radiation sources).

These 3 sources of radiation deliver low linear energy transfer (LET, which is defined by the rate of energy loss per unit length of track of the particle) radiations. The transient species produced by IRs in water have been well characterized in literature^{21,22}. In low-LET radiations, the first events appear in small widely separated spurs (10^{-16} - 10^{-10} s time scale) and the generated radicals are homogeneously spread into water at about 10^{-7} s after irradiation to yield aqueous electrons ($0.28 \mu\text{mol/J}$), hydrogen radicals (H^\bullet , $0.062 \mu\text{mol/J}$), hydroxyl radicals (HO^\bullet , $0.28 \mu\text{mol/J}$), hydrogen ($0.047 \mu\text{mol/J}$), hydrogen peroxide (H_2O_2 , $0.073 \mu\text{mol/J}$) and protons ($0.28 \mu\text{mol/J}$). Nevertheless, the radiochemical yield of HO^\bullet can be affected by the energy of the incident particle²³, and can be lowered when low-energy photons (1-100 keV) are used. To answer this question, we have quantified the amount of HO^\bullet generated by the X-ray generator (compared to the gamma-ray source) which deliver photons at energy affecting the HO^\bullet yield (cf. Supplementary Information section 9.1 p. 81, "Quantification of hydroxyl radicals and role of Gd", Figs. 59-63). The HO^\bullet production from the Kinetron and from the gamma-ray sources should be similar at the dose rate used ($0.28 \mu\text{mol/J}$)^{21,24}.

Indirect quantification of HO^\bullet was based on HO^\bullet scavenging by coumarin²⁵ and quantification of 7-hydroxycoumarin (7-OH-Coum) which is the only fluorescent product released from this scavenging reaction²⁶. 7-OH-Coum production proved to be proportional to HO^\bullet concentration and specific for HO^\bullet among other reactive oxygen species². This quantification method seems to be robust and adaptable^{27,28}. Nevertheless, it has to be noted that this is an indirect method of quantification which can only give access to the amount of HO^\bullet available to react with coumarin in our experimental conditions (temperature, concentrations, etc.). In the conditions used, coumarin reacts with HO^\bullet ~ 100 ns after the initial transfer of energy to water (rate constant of this scavenging reaction is $k = 1.05 \cdot 10^{10} \text{ L}\cdot\text{mol}^{-1}\cdot\text{s}^{-1}$)²⁹ which results in quantification of the homogeneously distributed HO^\bullet without interfering on the intratrack recombination of radicals occurring in the spurs.

The radiochemical yields (G -values) of HO^\bullet determined in our study were 0.200 and 0.280 $\mu\text{mol}/\text{J}$ upon X-ray and gamma-ray irradiations respectively (cf. Supplementary Information section 9.1 Fig. 63 when $[\text{Gd}^{3+}] = 0$, p. 85), which is in line with the reported yields for similar radiation sources and dose rates^{22,23}.

To conclude, the radiochemical yields of HO^\bullet ($G(\text{HO}^\bullet) \sim 100$ ns after irradiation) are 0.200, 0.280 and 0.280 $\mu\text{mol}/\text{J}$ for the X-ray, gamma-ray and electron sources respectively. Indeed, in this study, we showed that the amount of HO^\bullet was slightly lower from X-ray irradiation comparatively to the gamma-ray source, as expected from literature, even if the sources we used were all considered as low-LET radiations, with a similar radiolytic energy deposition. Interestingly, the activation yield of *cis*-GdAzo (G -value) was also slightly lower upon XR compared to GR and E at 2-5 Gy (cf. Fig. 3d in the main text and Fig. 47 p. 63 in the Supplementary Information section 5.4), even if this difference was lower than what could be expected from the $G(\text{HO}^\bullet)$ difference.

• The presence of Gd suggests to lead to higher isomerisation rate. If secondary electrons leading to OH radicals were the driving force then I would expect that it is not necessary to have Gd linked to the azobenzene. A Gd solution if well mixed should work similarly. Why was that not tested by the authors? An experiment in which the Gd concentration is varied would shine more light on the possible role that this element plays.

We would like to thank the reviewer for this very interesting suggestion. We, therefore, carried out the proposed experiment as detailed in Supplementary Information section 9.2 p. 85 (Impact of Gd on *cis*-Azo activation upon ionising radiations). However, we could not mix or stir the solutions during irradiation due to limitations of this type of ionising-radiation source, even if it would have been

interesting since HO^\bullet usually reacts at nearly diffusion-controlled rates²². As expected, a mix of *cis*-Azo and Gd^{3+} led to *cis*-Azo activation upon gamma-ray irradiation, and the higher the Gd^{3+} concentration was, the more efficient the *cis*-Azo activation was (Fig. N and Supplementary Information section 9.2 Fig. 64 p. 86).

Fig. N. Impact of the presence of Gd^{3+} for *cis*-Azo activation upon gamma-ray radiation. Absorbance difference (365 nm) of medium containing *cis*-GdAzo or *cis*-Azo (50 μM , H_2O) in the presence of Gd^{3+} (20, 50, 200, 500 and 2000 μM for C1 to C5 respectively, H_2O) before and after gamma-ray irradiation (triplicate). The experiment was repeated three times independently. The data are represented as the means \pm standard deviations. OD: optical density.

This experiment confirmed that the presence of Gd^{3+} was required to induce activation by ionizing radiation of this molecular system. Moreover, the activation was more efficient when the Gd^{3+} was close to the azobenzene moiety (cf. Fig. N, GdAzo vs Azo+Gd_C2, in which the same Gd^{3+} concentration was present).

• Furthermore can the authors determine how much more OH radicals are formed in the presence of Gd in comparison to samples without Gd? Why was that not attempted?

The indirect method for the quantification of HO• described above (using coumarin) has been used to determine the influence of the presence of Gd³⁺ on the radiochemical yield $G(\text{HO}^\bullet)$ upon X-ray and gamma-ray irradiations (Fig. O, Fig. 5d in the main text and Figs. 59-63 in the Supplementary Information section 9.1 p. 81). As already mentioned, when there is no Gd³⁺, $G(\text{HO}^\bullet)$ were 0.200 and 0.280 $\mu\text{mol}/\text{J}$ upon X-ray and gamma-ray irradiations respectively, which is similar to the values reported in literature^{22,23}. The yield of HO• was gradually increased in solutions with Gd³⁺ concentrations up to about 10-200 μM and then reached a plateau (25-500 μM) before decreasing for higher concentrations (500-2000 μM). The highest enhancement factors were 33% at 200 μM [Gd³⁺] and 20% at 500 μM [Gd³⁺] upon X-ray and gamma-ray irradiations respectively. The saturation and decrease of $G(\text{HO}^\bullet)$ at high Gd³⁺ concentrations could be explained by the increase in the probability of HO• recombination².

Fig. O. Determination of $G(\text{HO}^\bullet)$ at different concentrations of Gd³⁺. HO• production upon X-ray (XR) and gamma-ray (GR) in the presence of different concentrations of Gd³⁺ by considering a conversion yield of coumarin into 7-OH-Coum of 3.1% (from ref²). The experiments were repeated 5 times independently. The means \pm standard deviations are reported.

This study shows that the presence of Gd³⁺ in solution increases the radiochemical yield $G(\text{HO}^\bullet)$ upon ionizing radiation, even if this increase remains quite low. Noteworthy, this experiment quantified the HO• that diffused into the bulk solution and not the HO• initially generated into the spurs.

• To check the effect of OH radicals a chemical induction of OH radicals such as Fenton reactions can be applied. Why didn't the authors try that?

We thank the reviewer for this very interesting suggestion. As described just before, we performed this experiment for the new version of the manuscript (cf. Fig. L) and we observed that Fenton chemistry was able to activate *cis*-GdAzo, pointing out the role of HO[•] in this process (Figs. 5a, 5b in the main text and Supplementary Information section 8.2 p. 79).

• Finally, what about possible scintillation effects of Gd? I do not consider it very likely but I think that this should be discussed.

In the new version of the manuscript, several additional experiments have led to identify HO[•] as the main specie involved in the *cis*-GdAzo activation process. tBuOH, mannitol and EtOH should not have impacted *cis*-GdAzo activation with such a similar efficacy (cf. Fig. K) if scintillation effect was involved. Also, N₂O-saturation of solution should not have any effect on the *cis*-GdAzo activation (Fig. M). For clarity, this conclusion about the scintillation effect has been added in the main text (l. 316-319: "The central role of HO[•] led us to exclude some photon-mediated interactions in the *cis*-GdAzo activation process such as the Čerenkov effect (release of UV photons from accelerated charged particles such as electrons) and the scintillation effect (release of UV photons from gadolinium atoms)").

Finally, we decided to go further in the understanding of the activation mechanism. In the first version of the manuscript, we suggested a hypothesis for explaining the *cis*-GdAzo activation

mechanism, based on a catalytic effect (the oxidised *trans*-GdAzo^{••} was reduced by *cis*-GdAzo). We have tried to validate this hypothesis by investigating *cis*-GdAzo activation upon gamma-ray irradiation at different *cis*-GdAzo concentrations (20, 200, 500 and 1000 μ M, to be added to the 50 μ M concentration, as already assessed in the first version of the manuscript) (Fig. P). This additional study has been fully described in the Supplementary Information section 10 p. 87 (Investigation of activation mechanism by variation of GdAzo concentration).

Fig. P. Molecular activation of *cis*-Azo and *cis*-GdAzo at different concentrations upon gamma-ray radiation. Molecular activation of *cis*-GdAzo and *cis*-Azo (control molecule without Gd atom) at concentration of 20 μ M (a), 200 μ M (b), 500 μ M (c) and 1000 μ M (d) determined by HPLC and reported as the difference of *trans*-isomer proportion before and after gamma-ray (GR) irradiation ($n=3$). The means \pm standard deviations are reported. Two-way Anova (Bonferroni post-test) was used for statistical analyses (All vs GdAzo GR). *** $P < 0.001$. NI: non irradiated.

Thanks to those additional experiments, we could now determine the relation between the radiochemical yield (G -value) of *trans*-GdAzo and the initial concentration of *cis*-GdAzo. The linear relation observed (Fig Qa) shows that there is no catalytic effect for *cis*-GdAzo activation, which has the same efficacy for low and high *cis*-GdAzo concentration. Moreover, we observed that $G(\textit{trans}\text{-GdAzo})$ (*trans*-GdAzo production per unit energy) was higher for lower doses which reveals a larger loss of energy at higher doses (excess of HO^\bullet induces other non-specific reactions or recombination processes). This study allowed us to establish a predictive model to estimate $G(\textit{trans}\text{-GdAzo})$ from the initial *cis*-GdAzo concentration and the radiation dose, since $G(\textit{trans}\text{-GdAzo})$ follows a logarithmic decrease when the dose increases in the range 2-20 Gy (Fig Qd).

Fig. Q. Relation of G -values vs *cis*-GdAzo and *cis*-Azo concentrations upon gamma-ray radiation.

a, b, G -value ($\mu\text{mol/J}$) of the activation of *cis*-GdAzo (**a**) and *cis*-Azo (**b**) upon gamma ray determined by HPLC (molar amount of *trans*-GdAzo and *trans*-Azo activation corrected from thermal back relaxation) (3 independent experiments). **c**, Parameters obtained from linear regressions in **a** and **b**, using the least-squares method (Excel 2016). **d**, Linear regression to correlate $G(\textit{trans}\text{-GdAzo})/[\textit{cis}\text{-GdAzo}]$ to the irradiation doses. The relation $Y = -0.036 \cdot \ln(X) + 0.1304$ (with Y the slope of linear

regressions from \mathbf{a} in the unit $G\text{-value}/\mu\text{M GdAzo}$ and X the dose in Gy) was obtained with $r^2 = 0.9925$ (Excel 2016).

Thus, the bunch of experiments proposed in the new version of the manuscript sheds a new light on the underlying mechanism of *cis*-GdAzo activation upon ionising radiations. This process is not dependant on the type and energy of the incident particles (for low-LET radiation sources), even if the HO^\bullet amount released from radiation in the bulk solution slightly differs. The key role of HO^\bullet has been validated using several approaches (scavengers, gas saturation, chemical induction) which showed that HO^\bullet is the only specie responsible for *cis*-GdAzo activation. The presence of Gd^{3+} in solution increases the radiochemical yield $G(\text{HO}^\bullet)$ and leads to the activation of the control compound *cis*-Azo. Then, a relation to predict $G(\text{trans-GdAzo})$ after irradiation in the range 2-20 Gy has been established.

However, water radiolysis generates many events, including both reductive and oxidative processes and the understanding of these events are still intensively studied and remain prone to discussion (for instance, cf. ref¹ or ref³²). This study provides several elements and demonstrations that result in a better understanding of the implemented processes during *cis*-GdAzo activation and the authors acknowledge the excellent reviewer suggestions for mechanism clarification. However, we are conscious that the accurate mechanism underlying these complex processes will still require many efforts to be fully unlocked. Indeed, illustration of the complexity of ionising-radiation chemistry is revealed by the current discussions to fully understand the mechanisms of radioenhancement effect induced by metallic nanoparticles^{3,9}, even though this research field was born two decades ago⁴ and led to therapeutic approaches currently assessed in clinical trials^{5,6}.

References

1. Loh, Z.-H. *et al.* Observation of the fastest chemical processes in the radiolysis of water. *Science* **367**, 179-182 (2020).
2. Sicard-Roselli, C. *et al.* A New Mechanism for Hydroxyl Radical Production in Irradiated Nanoparticle Solutions. *Small* **10**, 3338-3346 (2014).
3. Kuncic, Z. & Lacombe, S. Nanoparticle radio-enhancement: principles, progress and application to cancer treatment. *Phys. Med. Biol.* **63**, 02TR01 (2018).
4. Hainfeld, J., Slatkin, D. & Smilowitz, H. The use of gold nanoparticles to enhance radiotherapy in mice. *Phys. Med. Biol.* **49**, N309-N315 (2004).
5. Bonvalot, S. *et al.* NBTXR3, a first-in-class radioenhancer hafnium oxide nanoparticle, plus radiotherapy versus radiotherapy alone in patients with locally advanced soft-tissue sarcoma (Act.In.Sarc): A multicentre, phase 2-3, randomised, controlled trial. *Lancet Oncol.* **20**, 1148-1159 (2019).
6. Verry, C. *et al.* Targeting brain metastases with ultrasmall theranostic nanoparticles, a first-in-human trial from an MRI perspective. *Sci. Adv.* **6**, eaay5279 (2020).
7. Song, G., Cheng, L., Chao, Y., Yang, K. & Liu, Z. Emerging nanotechnology and advanced materials for cancer radiation therapy. *Adv. Mater.* **29**, 1700996 (2017).
8. Ni, K. *et al.* Nanoscale metal-organic frameworks for x-ray activated in situ cancer vaccination. *Sci. Adv.* **6**, eabb5223 (2020).
9. Clement, S. *et al.* Mechanisms for tuning engineered nanomaterials to enhance radiation therapy of cancer. *Adv. Sci.* **7**, 2003584 (2020).
10. Fan, W. *et al.* Breaking the depth dependence by nanotechnology-enhanced X-ray-excited deep cancer theranostics. *Adv. Mater.* **31**, 1806381 (2019).
11. Chen, X., Song, J., Chen, X. & Yang, H. X-ray-activated nanosystems for theranostic applications. *Chem. Soc. Rev.* **48**, 3073-3101 (2019).
12. Sun, W., Zhou, Z., Pratz, G., Chen, X. & Chen, H. Nanoscintillator-Mediated X-Ray Induced Photodynamic Therapy for Deep-Seated Tumors: From Concept to Biomedical Applications. *Theranostics* **10**, 1296-1318 (2020).
13. Barosi, A. *et al.* Synthesis and activation of an iron oxide immobilized drug-mimicking reporter under conventional and pulsed X-ray irradiation conditions. *RSC Adv.* **10**, 3366-3370 (2020).
14. Wu, S.-Y. *et al.* Radiation-sensitive dendrimer-based drug delivery system. *Adv. Sci.* **5**, 1700339 (2018).
15. Fu, Q. *et al.* External-radiation-induced local hydroxylation enables remote release of functional molecules in tumors. *Angew. Chem. Int. Ed.* **59**, 21546-21552 (2020).
16. Zhang, F. *et al.* X-ray-triggered NO-released Bi-SNO nanoparticles: all-in-one nano-radiosensitizer with photothermal/gas therapy for enhanced radiotherapy. *Nanoscale* **12**, 19293-19307 (2020).

17. Fan, W. *et al.* Generic synthesis of small-sized hollow mesoporous organosilica nanoparticles for oxygen-independent X-ray-activated synergistic therapy. *Nat. Commun.* **10**, 1241 (2019).
18. Su, M., Guggenheim, K. G., Lien, J., Siegel, J. B. & Guo, T. X-ray-mediated release of molecules and engineered proteins from nanostructure surfaces. *ACS Appl. Mater. Interfaces* **10**, 31860-31864 (2018).
19. Zhou, Z. *et al.* Synchronous chemoradiation nanovesicles by X-Ray triggered cascade of drug release. *Angew. Chem. Int. Ed.* **57**, 8463-8467 (2018).
20. Deng, W. *et al.* Controlled gene and drug release from a liposomal delivery platform triggered by X-ray radiation. *Nat. Commun.* **9**, 2713 (2018).
21. Buxton, G. V. *Radiation Chemistry* (ed Mehran Mostafavi Mélanie Spothem-Maurizot, Thierry Douki & Jacqueline Belloni) Ch. 1, p. 3-16 (EDP Sciences, France, 2008).
22. Buxton, G. V., Greenstock, C. L., Phillips Helman, W. & Ross, A. B. Critical Review of rate constants for reactions of hydrated electrons, hydrogen atoms and hydroxyl radicals ($\cdot\text{OH}/\cdot\text{O}^-$) in Aqueous Solution. *J. Phys. Chem. Ref. Data* **17**, 513-886 (1988).
23. Fulford, J., Bonner, P., Goodhead, D. T., Hill, M. A. & O'Neill. Experimental determination of the dependence of OH radical yield on photon energy: A comparison with theoretical simulations. *J. Phys. Chem. A* **103**, 11345-11349 (1999).
24. Bobrowski, K. *Applications of Ionizing Radiation in Materials Processing* Vol. 1 (eds Yongxia Sun & Andrzej Grzegorz Chmielewski) Ch. 4, p. 81-116 (Institute of Nuclear Chemistry and Technology, Warszawa, 2017).
25. Singh, T. S., Madhava Rao, B. S., Mohan, H. & Mittal, J. P. A pulse radiolysis study of coumarin and its derivatives. *J. Photochem. Photobiol.* **153**, 163-171 (2002).
26. Louit, G. *et al.* The reaction of coumarin with the OH radical revisited: hydroxylation product analysis determined by fluorescence and chromatography. *Radiat. Phys. Chem.* **72**, 119-124 (2005).
27. Louit, G. *et al.* Determination of hydroxyl rate constants by a high-throughput fluorimetric assay: towards a unified reactivity scale for antioxidants. *Analyst* **134**, 250-255 (2009).
28. Náfrádi, M. *et al.* Application of coumarin and coumarin-3-carboxylic acid for the determination of hydroxyl radicals during different advanced oxidation processes. *Radiat. Phys. Chem.* **170**, 108610 (2020).
29. Brun, E., Girard, H. A., Arnault, J.-C., Mermoux, M. & Sicard-Roselli, C. Hydrogen plasma treated nanodiamonds lead to an overproduction of hydroxyl radicals and solvated electrons in solution under ionizing radiation. *Carbon* **162**, 510-518 (2020).
30. Frelon, S., Douki, T., Favier, A. & Cadet, J. Comparative study of base damage induced by gamma radiation and Fenton reaction in isolated DNA. *J. Chem. Soc., Perkin Trans. 1*, 2866-2870 (2002).
31. Miller, C. J., Rose, A. L. & Waite, T. D. Importance of iron complexation for Fenton-mediated hydroxyl radical production at circumneutral pH. *Front. Mar. Sci.* **3** (2016).
32. Svoboda, V. *et al.* Real-time observation of water radiolysis and hydrated electron formation induced by extreme-ultraviolet pulses. *Sci. Adv.* **6**, eaaz0385-eaaz0385 (2020).

33. Nagle, J. F. & Tristram-Nagle, S. Structure of lipid bilayers. *Biochim. Biophys. Acta* **1469**, 159-195 (2000).
34. Walev, I. *et al.* Delivery of proteins into living cells by reversible membrane permeabilization with streptolysin-O. *Proc. Natl. Acad. Sci. USA* **98**, 3185-3190 (2001).
35. Rosenberg, M., Azevedo, N. F. & Ivask, A. Propidium iodide staining underestimates viability of adherent bacterial cells. *Sci. Rep.* **9**, 6483 (2019).
36. Kirchoff, C. & Cypionka, H. Propidium ion enters viable cells with high membrane potential during live-dead staining. *J. Microbiol. Methods* **142**, 79-82 (2017).
37. Shi, L. *et al.* Limits of propidium iodide as a cell viability indicator for environmental bacteria. *Cytometry* **71A**, 592-598 (2007).
38. Hüll, K., Morstein, J. & Trauner, D. In vivo photopharmacology. *Chem. Rev.* **118**, 10710-10747 (2018).
39. Chithrani, D. *et al.* Gold nanoparticles as radiation sensitizers in cancer therapy. *Radiat. Res.* **173**, 719 - 728 (2010).
40. Li, S. *et al.* Platinum nanoparticles: an exquisite tool to overcome radioresistance. *Cancer Nano.* **8**, 4 (2017).
41. Lux, F. *et al.* AGuIX® from bench to bedside—Transfer of an ultrasmall theranostic gadolinium-based nanoparticle to clinical medicine. *Brit. J. Radiol.* **91**, 20180365 (2018).
42. Bort, G. *et al.* EPR-mediated tumor targeting using ultrasmall-hybrid nanoparticles: From animal to human with theranostic AGuIX nanoparticles. *Theranostics* **10**, 1319-1331 (2020).
43. Singh, A., Koroll, G. W. & Cundall, R. B. Pulse radiolysis of aqueous solutions of sodium azide: Reactions of azide radical with tryptophan and tyrosine. *Radiat. Phys. Chem.* **19**, 137-146 (1982).

REVIEWERS' COMMENTS

Reviewer #1 (Remarks to the Author):

The manuscript is now much improved.

I would recommend publication only in case that (some of) the limitations of the system as depicted through my comments in the previous round (see below) are clearly stated within the manuscript for the reader's benefit :

6. it seems that the GdAzo-caused enhancement in cell killing was quite low without even taking the error bars into account. If one applies simple mathematics to the data presented in Fig. 4h and crude numbers taken from the presented figure in then only 10% enhancement can be obtained: $(\text{NIGdAzo} - \text{GRGdAzo}) - (\text{Ctrl} - \text{GR}) = 80 - 25 - (100 - 55) = 55 - 45 = 10\%$. This is too low a synergy also presented in the showcase results. Perhaps it would be a good idea for the authors to present their results in different experiments so the median gain from the use of cisGdAzo can become clearer.

The new paragraph supplied by the authors in response to my comment no 7: "It has to be noted that the thermal half-life of cis-GdAzo is well adapted for in vitro experiments, however it should be increased to several days or weeks for in vivo investigation" (l. 154-156 in the main text)

should be changed to:

"It has to be noted that while the thermal half-life of cis-GdAzo is suitable for in vitro experiment, for any in vivo work new systems need be developed, with a thermal half-lives of several days to weeks " (l. 154-156 in the main text)
or something equivalent.

Reviewer #3 (Remarks to the Author):

I am fully satisfied with the new experimental data and I recommend publishing. The paper is very interesting!

Response to reviewers' comments - NCOMMS-20-22126-A**Reviewer #1 (Remarks to the Author):**

The manuscript is now much improved.

I would recommend publication only in case that (some of) the limitations of the system as depicted through my comments in the previous round (see below) are clearly stated within the manuscript for the reader's benefit :

6. it seems that the GdAzo-caused enhancement in cell killing was quite low without even taking the error bars into account. If one applies simple mathematics to the data presented in Fig. 4h and crude numbers taken from the presented figure in then only 10% enhancement can be obtained: $(\text{NIGdAzo} - \text{GRGdAzo}) - (\text{Ctrl} - \text{GR}) = 80 - 25 - (100 - 55) = 55 - 45 = 10\%$. This is too low a synergy also presented in the showcase results. Perhaps it would be a good idea for the authors to present their results in different experiments so the median gain from the use of *cis*-GdAzo can become clearer.

As mentioned in our previous answer to the reviewer, we agree that we cannot conclude to a synergistic effect according to the data presented here (10% is small and we do not have carried out a specific study to show it such as the determination of combination index), and such a conclusion has never been claimed in the text. Nevertheless, the increase of toxicity for the combination of *cis*-GdAzo and ionising radiation was observed, and bring out interesting outcomes, such as a nearly complete cancer-cell death at high concentration *cis*-GdAzo concentration (remaining of 2.8% living cells).

To clarify this point to the reader's benefit, this sentence has been introduced: "Even if the synergistic cytotoxic effect of *cis*-**GdAzo** and IR will need to be improved, this approach brought out that nearly complete cancer-cell death can be achieved at high concentration of *cis*-**GdAzo** upon IR (remaining of 2.8% living cells)." (l. 571-573 in the main text).

The new paragraph supplied by the authors in response to my comment no 7: "It has to be noted that the thermal half-life of *cis*-GdAzo is well adapted for *in vitro* experiments, however it should be increased to several days or weeks for *in vivo* investigation" (l. 154-156 in the main text)

should be changed to:

"It has to be noted that while the thermal half-life of *cis*-GdAzo is suitable for *in vitro* experiment, for any *in vivo* work new systems need be developed, with a thermal half-lives of several days to weeks " (l. 154-156 in the main text) or something equivalent.

This sentence has been replaced by: "It has to be noted that while the thermal half-life of *cis*-**GdAzo** is suitable for *in vitro* experiment, for any *in vivo* work, other systems need be developed, with thermal half-lives of several days to weeks" (l. 157-159 in the main text).

Reviewer #3 (Remarks to the Author):

I am fully satisfied with the new experimental data and I recommend publishing. The paper is very interesting!

The authors thank very much the reviewer for his enthusiastic comment and for having provided interesting advices to improve our work.